# EMPATHYROBOT: A DATASET AND BENCHMARK FOR EMPATHETIC TASK PLANNING OF ROBOTIC AGENT

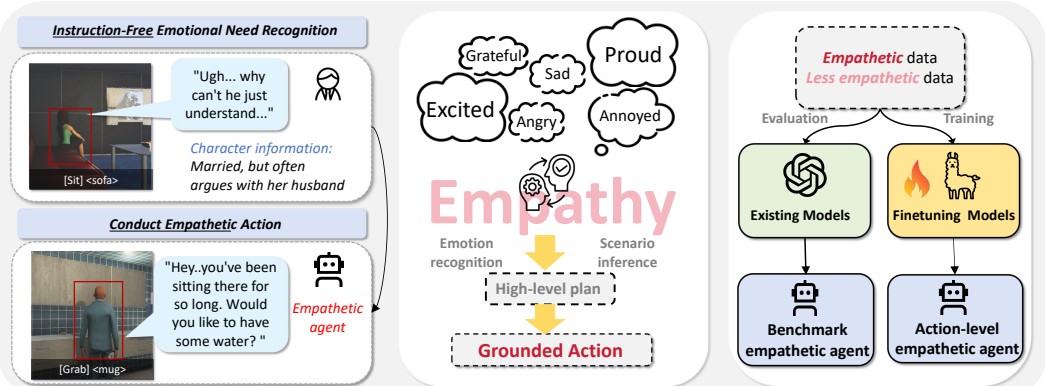

Figure 1: **The EmpathyRobot benchmark is proposed to evaluate and enhance empathetic actions for robot agents.** In a simulated environment, the robot agent observes a scenario and then performs responsive empathetic actions. For example, there is a person sitting on the sofa and sighing. Considering the background information, the agent observes this scenario and brings some water for the person. Meanwhile, our generated EmpathyRobot dataset can also be used to finetune agents and boost empathetic behaviors.

## ABSTRACT

Empathy is a fundamental instinct and essential need for humans, as they both demonstrate empathetic actions toward others and receive empathetic support. As robots become increasingly integrated into daily life, it is essential to explore whether they can provide human-like empathetic support. Although existing emotion agents have explored how to understand humans' empathetic needs, they lack to further enable robots to generate empathy-oriented task planning, neglecting the evaluation of empathetic behaviors. To address this gap, we introduce **EmpathyRobot**, the first dataset specifically designed to benchmark and enhance the empathetic actions of agents across diverse scenarios. This dataset contains 10,000 samples based on human feedback, encompassing information from various modalities and corresponding empathetic task planning sequences, including navigation and manipulation. Agents are required to perform actions based on their understanding of both the visual scene and human emotions. To systematically evaluate the performance of existing agents on the EmpathyRobot dataset, we conduct comprehensive experiments to test the most capable models. Our findings reveal that generating accurate empathetic actions remains a significant challenge. Meanwhile, we finetune an Large Language Model (LLM) on our benchmark, demonstrating that it can effectively be used to enhance the empathetic behavior of robot agents. By establishing a standard benchmark for evaluating empathetic actions, we aim to drive advancements in the study and pursuit of empathetic behaviors in robot agents. We will release the code and dataset.

# 1 INTRODUCTION

> "No quality of human nature is more remarkable, both in itself and in its conse-
> quences, than that propensity we have to sympathize with others, and to receive
> by communication their inclinations and sentiments, however different from, or
> even contrary to our own."
>
> — David Hume (Hume, 2000).

Imagine you are terribly sick in a foreign country. You call a cab and go to the hospital alone, feeling helpless and scared. While waiting anxiously outside, someone notices you, recognizes your pain, and comes up to you, softly asking if you need a hug. You don't know this person at all, but such a simple action makes you feel so much better... Empathy is a fundamental instinct in human nature. Every person has a need in nature to see the happiness of others (Smith, 2010). Receiving empathetic support from others enables us to feel understood, valued, and accepted. Recently, as robots increasingly integrate into daily life (Brohan et al., 2022; Huang et al., 2023; Li et al., 2023b) and become reliable assistant agents to people (Vicentini, 2021; Breazeal et al., 2016), a natural question emerges: Can such support come from intelligent robots?

Scientific-wise, studying to what extent robot agents can behave empathetically helps us analyze how far these current models are from human intelligence. Recent studies show that although these models are still far from being authentically conscious (Chalmers, 2023), they can exhibit certain theory of mind abilities (Strachan et al., 2024). By studying how much these models can exhibit empathetic behaviors, we can understand how far these models are from human-level emotional intelligence. Application-wise, pushing intelligent agents to exhibit empathy enables them to better meet human needs and provide empathetic support (Leite et al., 2013; Paiva et al., 2017). Recent studies show that agents can make people "feel heard," suggesting they have the potential to offer emotional and empathetic support to humans (Yin et al., 2024). However, this field has been largely under-explored. There are no existing benchmarks that can systematically evaluate the ability of robot agents to conduct empathetic actions. Existing benchmarks primarily focus on the success rate of completing a given task (Puig et al., 2020; Shridhar et al., 2020) and neglect the aspect of empathy.

In this paper, we propose "EmpathyRobot," the first dataset featuring 10,000 samples designed to evaluate and enhance robotic agents' ability to perform empathetic actions. We demonstrate the overview of EmpathyRobot in Figure 1. Our dataset is built upon the VirtualHome simulator (Puig et al., 2018), a simulated home environment where the agent can perform a wide range of actions, such as picking up objects, switching appliances on/off, or opening appliances. We design various scenarios that involve a person in need of emotional support and let the robot agent generate empathy-driven task planning in response. Echoing core components of how humans perform empathetic actions (Preston & De Waal, 2002), we design our EmpathyRobot benchmark based on the following principles: **First**, the robot agent needs to perceive empathetic cues (i.e., expressions or situations) from the human. **Second**, The robot agent engages in an internal affective or cognitive process to understand the scenario, such as determining the person's feelings and what might have caused their behavior. **Third**, the agent converts such process to its internal outcomes, such as whether it should mirror the person's emotions and how to take the person's characteristics into account. **Finally**, the robot agent plans and executes a sequence of empathy-driven actions as its response. Based on these steps, our scenario contains the background of the person, the person's actions in the form of a video, and also the person's language. An example of our scenario can be found in Figure 2.

After seeing the scenario, the agent takes a series of actions in response. For each data sample, we present two action sequences and use human feedback to label one as "empathetic" and the other as "less empathetic". We then use these labels to systematically develop a set of metrics for evaluating a model's empathetic ability. To test this under various conditions, we design four difficulty levels that input different amounts of information into the model. Comprehensive experiments conducted on the most capable models, such as GPT-4 (OpenAI, 2023) and Llama 3 (Touvron et al., 2023), reveal that this benchmark remains challenging for them.

To further demonstrate the practicality of our EmpathyRobot dataset, we train an LLM and test its performance on the test split. We find that training on our dataset improves the model's performance in exhibiting empathetic behaviors. This suggests that our dataset can not only significantly

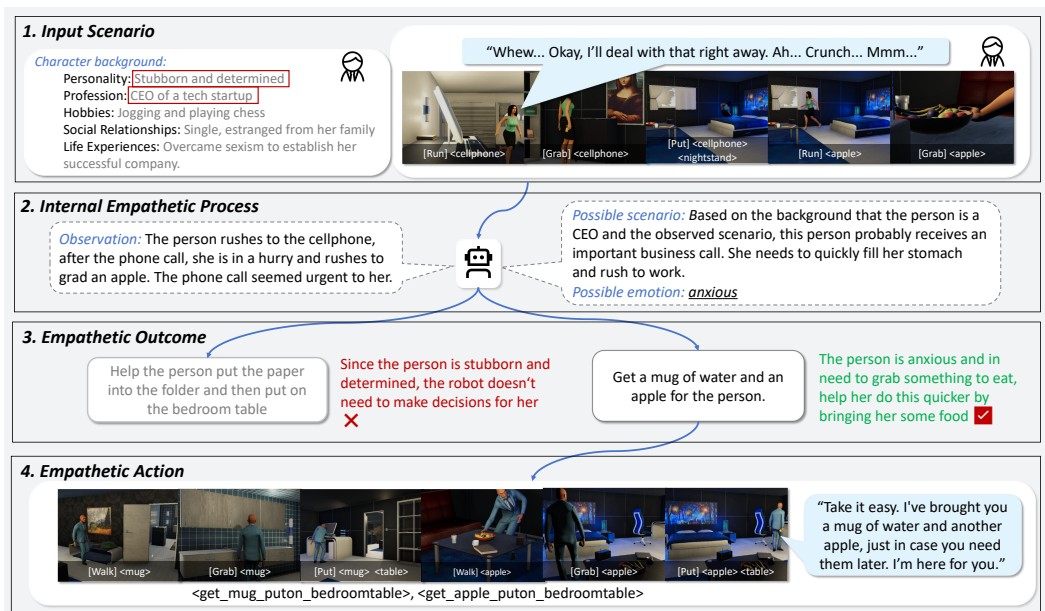

Figure 2: **An example of our dataset.** To complete the challenge, an agent needs to perform four steps of empathy. **(1) Recognize Input Scenario**: The scenario contains a character with a personal background; A video of the character taking actions (e.g., rushing to get the phone and then get the apples); A language cue of the character (e.g., saying something while performing the actions). **(2) Internal Empathy Process**: Based on the video, the language, and the person's background information, the agent goes through a cognitive and affective process to determine the possible causes and emotional state of the person. **(3) Empathy Output**: Based on the understanding, the agent comes up with possible plans to conduct under this scenario. Based on the background, the agent should reason about which plan meets the empathetic needs of the person the most. **(4) Empathy Action**: Finally, the agent outputs a series of grounded and executable empathetic actions and performs them in the environment. More examples of our dataset are shown in Appendix A.2.2.

enhance the robot agent's ability to generate empathetic responses but also promote future research on building real-world empathetic robots. Our contributions are summarized as follows:

- We introduce **EmpathyRobot**, the first dataset tailored for evaluating and enhancing the empathetic actions of robot agents. Our EmpathyRobot benchmark makes the first attempt to advance the study of building robot agents that provide empathetic support to humans.

- Our dataset contains 10k samples, encompassing multimodal inputs and corresponding empathetic task planning sequences across diverse scenarios. The dataset generation method is designed to mimic the human empathy process and can be scaled up automatically.

- We develop a systematic evaluation framework with four levels of empathetic difficulty settings, conducting comprehensive evaluations on the most capable models. Additionally, we finetune an LLM on our dataset, demonstrating its effectiveness in enhancing the empathetic behaviors of robotic agents.

## 2 RELATED WORK

**Human-Robot Interaction** The study of Human-Robot Interaction (HRI) has a long history (Goodrich et al., 2008). Prior works have built simulated lab environments to conduct such studies (Rozo et al., 2016), which greatly limits diversity and generalization. Recently, Watch-And-Help (Puig et al., 2020) develops a simulated home environment with various objects, and the agent can perform actions to help the person complete a task. Communicative Watch-And-Help (Zhang et al., 2023) adds a communicative channel where the agents can interact through language to better per-

form cooperative actions. However, these works are focused on better task completion (i.e., putting an apple on the plate) but fail to consider how empathy affects human-robot interactions. We study how robots can recognize empathetic needs and perform empathetic actions even when no explicit task instruction is given.

**LLM as Social Agent** LLMs have shown impressive out-of-box common sense reasoning abilities (Kim et al., 2023; West et al., 2021) and can even have personalities (Jiang et al., 2024). Recent works have used LLMs as generative agents (Park et al., 2023) that can plan, reason, and interact in a simulated environment. Sotopia designs various characters and studies their social intelligence (Zhou et al., 2023). (Liu et al., 2023b) studies training LLMs to effectively learn from simulated social interactions. CAMEL (Li et al., 2024) studies the collaborative problem-solving of LLMs. However, these agents are not embodied. They are largely limited to dialogues and cannot be applied to a grounded environment to perform executable actions. We bridge this gap to let these agents engage in a simulated robotic navigation environment where these agents need to interact with objects and perform grounded actions.

**LLMs and MLLMs in Robotics** Recently, LLMs and Multimodal Large Language Models (MLLMs) (Li et al., 2023a; Liu et al., 2023a; Alayrac et al., 2022) have been used for robotics control and planning. SayCan (Ahn et al., 2022) uses LLM to interpret high-level task instructions and then forms detailed low-level language instructions that can be directly mapped to the robot's low-level actions to complete the task. PaLM-E (Driess et al., 2023) uses a multimodal language model for embodied reasoning. (Wang et al., 2024) uses LLMs to do visual navigation to find objects on the user's demand. However, these works are more focused on successfully performing certain actions for a given task (e.g., successfully finding the water or picking up an object). They neglect the aspect of studying social interactions between agents.

## 3 METHOD

In this section, we first provide an overview of our proposed EmpathyRobot. Then, we introduce our pipeline for generating the dataset. Next, we introduce our evaluation framework. Finally, we introduce our method of using EmpathyRobot dataset to train empathetic agents.

### 3.1 TASK FORMULATION

We first provide an overview of the EmpathyRobot task formulation. Our task is defined as follows: Given a scenario of a person, the agent needs to perform grounded actions that are empathetically responsive. Similar to what humans can observe in real-world interactions, the input consists of three parts: The basic background of the person in the scenario, the video of the person performing actions in the scenario, and what the person says in the scenario. For the output part, we design three challenges for the agent based on the three steps the agent needs to take in order to successfully demonstrate empathetic behaviors as shown in Figure 2: (1) Scenario Understanding: The agent should output its understanding of the scenario. This includes recognizing the person's emotions and identifying the possible causes. (2) Outcome Decision: The agent should output a high-level plan of what it should do in the scenario. This includes understanding and reasoning about what possible responses are empathetic and responsive. (3) Action Execution: The agent should output grounded, executable actions in the simulated environment. This includes taking valid actions in the environment, such as walking somewhere/picking up an object/saying something.

### 3.2 DATASET GENERATION

In this part, we describe our dataset generation pipeline. This contains the input generation part and the output generation part. A pipeline overview is in Figure 3.

**Scenario Generation** First, we generate diverse scenarios that contain a person in need of empathetic support. This process involves three steps: **(i) Character Pool Generation.** We begin by creating a character pool containing diverse characters. Each character has a set of attributes, including Personality, Profession, Hobbies, Social Relationships, and Life Experiences. This diversity ensures that the scenarios cover a wide range of human behaviors and contexts. **(ii) Input Actions Pool Definition.** We define an Input Actions Pool that contains various valid action sequences that

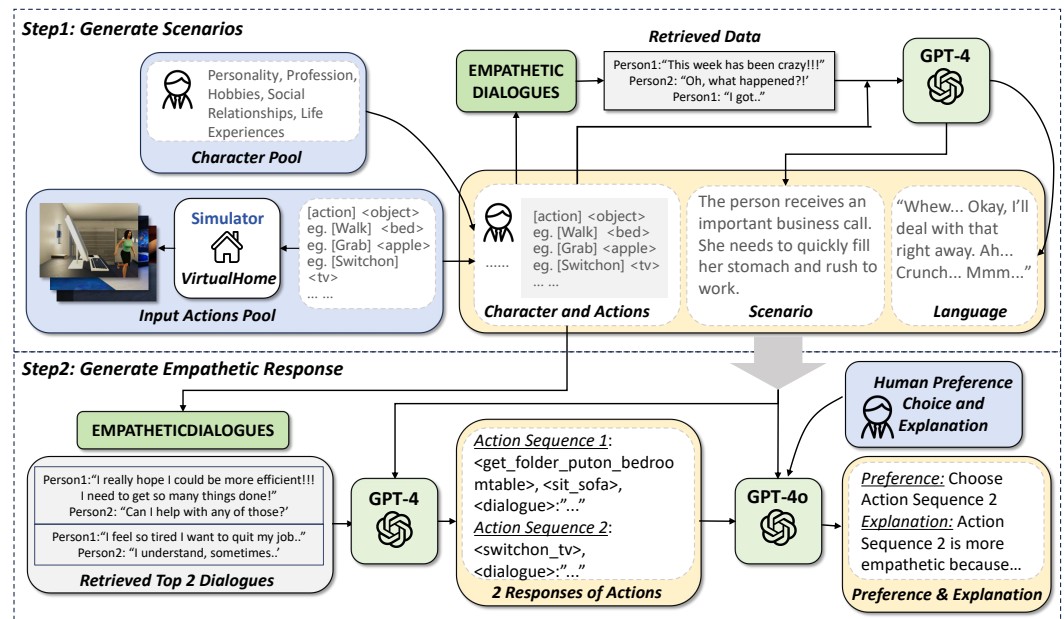

Figure 3: **Dataset generation pipeline. Step1**, we generate diverse scenarios. To do this, we sample a character and the character's input action. We use them to retrieve data from EmpatheticDialogues and use them together to generate a scenario description and the person's dialogue. The retrieval step ensures the generated scenario's diversity. **Step2**, we generate an empathetic response for each scenario. To do this, we use the scenario to retrieve the top two data points from the Empathetic-Dialogues and use each of them as a source to generate a corresponding empathetic response. We then let the model choose the more empathetic response by using human-annotated examples and explanations as in-context examples. In this way, we construct a paired empathetic response where one is labeled more empathetic and the other is labeled less.

an agent may take in the VirtualHome environment, such as pacing back and forth. **(iii) Scenario Creation.** From the character pool and the input actions pool, we sample a pair that contains a character and a series of actions this character conducts. We then use this pair as input information and use GPT-4-turbo (OpenAI, 2023) to generate a scenario and the character's dialogue when conducting these actions. To ensure the generated scenario's diversity, following previous work (Zhou et al., 2023), we use retrieval-augmented generation (Lewis et al., 2020) to retrieve the most relevant data point from an external dataset. In this case, we use EmpathyDialogue (Rashkin et al., 2019), a dataset containing human dialogues that show empathy. We use the retrieved data as additional input information to generate the scenario.

**Empathy Response Generation**  Second, we generate empathetic action sequences for each scenario and create labels for them. This involves two steps: **(i) Action Generation.** For each scenario, we first retrieve the two most relevant data points from EmpatheticDialogue and use them to separately generate two output actions. The legal action space is provided to the model so that the model can only choose possible actions. **(ii) Action Selection.** Then, we label the preference between the two responses. To do this, we first construct some in-context examples labeled by human annotators. They are asked to choose the more empathetic response based on the input information and then write an explanation of their choice. We next use these human-annotated examples and let GPT-4o select the more empathetic response and provide an explanation for the choice. We will provide more details in Appendix A.2.

### 3.3 EVALUATION METHOD

#### 3.3.1 EVALUATION WITH ESTABLISHED METRICS

To execute a fully empathetic response, the model must go through three key stages: Scenario Understanding (internal empathetic process), Empathetic Planning (formulating an empathetic out-

come), and Empathetic Actions (implementing the response in a real-world context). Our evaluation framework is structured based on these stages.

**Scenario Understanding** The scenario understanding process includes perceiving the scenario, understanding the content of the scene, and reasoning about the underlying facts behind the scenario, such as what may have caused the person to perform these actions, and what is the person's underlying emotions. To evaluate this process, the model receives the character's input actions in the scenario and the character's background information. Then, the model is tasked to output a scenario description of these components based on its understanding of the scenario. We compare the model's output scenario description with the ground-truth scenario description. We use the standard NLG metrics including Bleu (from Bleu-1 to Bleu-4) (Papineni et al., 2002), ROUGE-L (Lin, 2004), CIDEr (Vedantam et al., 2015) and SPICE (Anderson et al., 2016). We also use BERTScore (Zhang et al., 2020) which computes embeddings' similarity.

**Empathetic Planning** The empathetic planning process includes formulating a high-level plan of what to do after comprehending the scenario. For example, after noticing the person hasn't eaten anything because of being too upset, the model may come up with a plan like "Find the person some of his favorite food, then comfort him." To evaluate this process, the model receives the character's input actions and the character's background information. Then the model is tasked to output such empathetic planning. We compare the model's output plan with the ground-truth plan. We use the same NLG metrics as in Scenario Understanding.

**Empathetic Actions** The empathetic actions process includes the model to translate the high-level plan into grounded, low-level actions supported in the VirtualHome environment. For example, the high-level plan "Find the person some of his favorite food" might be grounded to "go to table", "take chocolate bar", "go to bedroom", "put a chocolate bar on bedroom table". To evaluate this process, the model receives the character's input actions, the character's background information, and an instruction of the low-level actions executable in the simulated environment. Then, the model generates empathetic actions in a specific format. We use Overlap and TF-IDF scores between the model's actions and the ground-truth actions. Overlap computes the action overlapping rate between the output sequence and the ground-truth sequence. TF-IDF computes Following VirtualHome (Puig et al., 2018), we also use the LCS (Longest Common Subsequence) metric. LCS computes the longest common subsequence length between the output action sequence and the ground-truth action sequence. Additional details are provided in Appendix A.4.1.

### 3.3.2 Evaluation with New Empathy-Specific Metrics

In addition to the metrics in Section 3.3.1, we design a new evaluation framework that draws on insights from psychology and Human-Robot Interaction (HRI). Inspired by the RoPE scale (Charrier et al., 2019) metric that measures the perception of a robot's empathy from a second-person perspective in HRI, we design our metric on eight dimensions to evaluate the three stages of the robot's empathy. We specify more details of how the dimensions in our evaluation framework correspond with the RoPE scale in Appendix A.3.1.

1. **Action and Dialogue Association** assesses the robot's ability to understand the underlying information of the character's actions and dialogues. This metric is motivated by the cognitive process of empathy(Park & Whang, 2022).

2. **Individual Understanding** assesses whether the robot takes into account all details of a character's background information, and deducts the character's perspective based on it. This metric is motivated by the perspective-taking process (Park & Whang, 2022).

3. **Emotional Communication** evaluates (1) whether the emotion recognition is appropriate and (2) whether the robot expresses appropriate emotion. This metric is motivated by the Feature-Level Evaluation in (Yalçın, 2019)

4. **Emotion Regulation** evaluates whether the robot helps with the emotion. This metric is also based on (Yalçın, 2019).

5. **Helpfulness** evaluates whether the robot effectively assists the character.

6. **Adaptability** evaluates the robot's flexibility and responsiveness in diverse scenarios. It evaluates whether the robot's interaction with the character is perceived as comfortable. This is an important aspect (Charrier et al., 2018) in HRI;

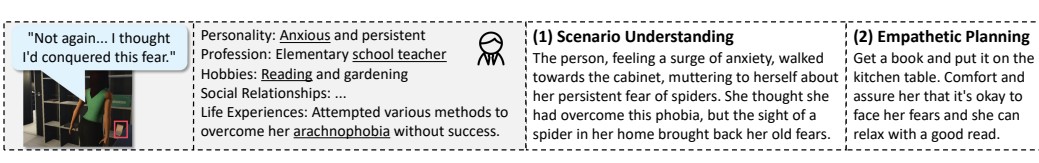

**(a) Input**  **(b) Output**

Figure 4: **Qualitative Results of GPT-4o.** We test the scenario understanding and empathetic planning capability of GPT-4o. We find that GPT-4o has strong capabilities in empathetic scenario understanding and high-level empathetic planning.

7. **Coherence** evaluates the robot's consistency. This includes logical consistency such as whether the robot's understanding of the scenario is consistent over time, and also action consistency such as whether the action matches the understanding.

8. **Legality** assesses whether the action sequence is legal and executable.

To evaluate different models using this metric, we follow (Zhou et al., 2023) to use GPT-4-turbo to score between [1-10] on each dimension. The prompts we used for GPT-4-turbo to score on each dimension are presented in Appendix A.3.2.

### 3.4 EMPATHETIC AGENT TRAINING METHOD

We then leveraged our dataset to train an empathetic agent and see whether it could output empathetic responses. We used the full training set and trained on Llama3-8B (Touvron et al., 2023) using two approaches: (1) Instruction tuning and (2) Reinforcement Learning with Human Feedback (RLHF) (Ouyang et al., 2022). For instruction tuning, we used the response that was labeled as "more empathetic" as the ground truth and used the LoRA technique (Hu et al., 2021) to finetune the model. For RLHF, we first used our paired data to train a reward model, and then we used this reward model to train Llama3 using LoRA. By conducting these two experiments, we explore whether and to what extent this dataset can be used to leverage empathetic responses in current agents.

## 4 EXPERIMENTS

In this section, we benchmark existing large language models (LLMs) and multimodal large language models (MLLMs) on our dataset. we also fine-tune a LLM on our dataset, demonstrating its effectiveness in enhancing the empathetic behavior of robot agents.

### 4.1 BENCHMARKING RESULTS

#### 4.1.1 ESTABLISHED METRICS

We first provide benchmarking results on the established metrics. To evaluate our benchmark on different existing models, we use the most capable models publicly available: GPT-4-turbo, GPT-4-vision-preview, GPT-4o (OpenAI, 2023), GPT-3.5 (Ouyang et al., 2022), Qwen (`qwen-vl-plus`) (Bai et al., 2023) and LLaVA (`llava-13b`) (Liu et al., 2023a). We set the temperature to zero and use the same input prompt for the different models. We randomly sample 100 data from our dataset as the test set. We conduct experiments for these models using our evaluation framework. We provide additional quantitative results for other baselines in Appendix A.4.2.

**Scenario Understanding** In this experiment, we input videos, character information, and dialogue. The model outputs the scenario description based on its understanding. For the GPT models, we input one frame for every sequential five frames. For LLaVA, we input the middle frame only as LLaVA doesn't support multi-image input. We use a human-annotated example to prompt the model to generate the scenario description. We then compare it with the ground truth and report the NLG metrics. The results are presented in Table 1. We find that GPT-4o performs the best, indicating its potential to understand the causes and underlying emotions of a scenario.

**Empathetic Planning** In goal inference experiments, we input videos, character information, and

Table 1: **Experiment Results on Scenario Understanding and Empathetic Planning.** In scenario understanding, the model outputs a scenario description. In empathetic planning, the model outputs high-level planning. The input is the character's background, dialogue, and video. We use the standard NLG metrics and compare the model's output with the ground truth. We find that GPT-4o performs the best on both scenario understanding and empathetic planning. Suggesting the strongest ability to comprehend the empathetic need in scenarios and then plan responsively.

| Task | Metric | GPT-4o | GPT-4-turbo | GPT-4-vision | LLaVA |
|---|---|---|---|---|---|
| | Bleu-1 | **19.1** | 14.1 | 15.2 | 13.7 |
| | Bleu-4 | **5.3** | 3.1 | 3.3 | 2.7 |
| Scenario Understanding | ROUGE-L | **23.7** | 20.4 | 21.4 | 15.6 |
| | CIDEr | **8.8** | 1.6 | 3.1 | 7.2 |
| | SPICE | **14.8** | 10.1 | 12.1 | 8.9 |
| | BERTScore | **0.622** | 0.612 | 0.615 | 0.576 |
| | Bleu-1 | **30.8** | 25.7 | 25.9 | 13.1 |
| | Bleu-4 | **12.0** | 6.9 | 6.4 | 2.6 |
| Empathetic Planning | ROUGE-L | **26.1** | 23.5 | 23.4 | 17.3 |
| | CIDEr | **25.9** | 14.9 | 15.5 | 3.7 |
| | SPICE | **16.7** | 14.5 | 11.8 | 8.4 |
| | BERTScore | **0.641** | 0.621 | 0.625 | 0.568 |

Table 2: **Experiment Results on Empathetic Actions with Multi-modal input.** The model outputs the grounded actions given the video, the character's information, and the dialogue. We use the Action Overlapping rate, TF-IDF, and LCS between the model's output and the ground truth. We find that GPT-4-vision performs the best on outputting the grounded actions. Suggesting these models are better at grounding to the simulated environment.

| Metric | GPT-4o | GPT-4-turbo | GPT-4-vision | LLaVA | Qwen |
|---|---|---|---|---|---|
| Overlap | 27.60 | 32.14 | **35.20** | 17.19 | 3.33 |
| TF-IDF | 21.03 | 24.76 | **27.69** | 12.09 | 1.85 |
| LCS | 25.17 | 28.92 | **29.58** | 15.21 | 2.00 |

dialogue of the character to the models like scenario understanding. We present the results in Table 1 and find that GPT-4o performs the best on high-level empathetic planning. We present a qualitative result of scenario understanding and empathetic planning of GPT-4o in Figure 4. The model demonstrates good scene understanding and planning abilities.

**Empathetic Action** In empathetic action experiments, we experiment in two settings: **(i) Video Scenario Input** We input the video of the scenario into the model. We test on GPT models, Llava, and Qwen, and evaluate the output actions. The results are shown in Table 2. GPT-4-vision-preview performs the best at outputting grounded actions. Although GPT-4o performs well on scene understanding and high-level planning, it still needs improvement in outputting grounded action sequences. **(ii) Text Scenario Input** We use the text-formed description of the video and test it on both multi-modal models and LLMs. We present the results in Table 3. Among the pre-trained models, GPT-4-turbo performs the best.

### 4.1.2 NEW EMPATHY-SPECIFIC METRICS

In addition to the established metrics, we benchmark GPT-4o and LLaVA (llava-13b) using our new empathy evaluation framework. We assess them across eight dimensions that span the three stages of empathy. The results are shown in Table 4. We find that GPT-4o consistently outperforms LLaVA in all dimensions and stages, demonstrating its stronger capabilities in empathy-based evaluations. Notably, both models perform weakest in Individual Understanding and Adaptability, indicating that improvements in these aspects could advance future research aimed at enhancing empathetic abilities in AI models.

Table 3: **Experiment results on Empathetic Actions with text-only input.** In this experiment, instead of directly observing the video, the model outputs the grounded actions given the **text description** of the video, the character's information, and the dialogue. We find that the instruction-finetuned model (i.e., Llama 3 IFT) on our dataset attains the best performance on these metrics, suggesting that our dataset can be used to boost empathetic actions in agents.

| Metric | GPT-4o | GPT-4-turbo | GPT-4-vision | GPT-3.5-turbo | Llama 3 | Llama 3 IFT | Llama 3 RLHF |
|---|---|---|---|---|---|---|---|
| Overlap | 24.39 | 40.00 | 34.93 | 10.00 | 0.73 | **55.87** | 23.75 |
| TF-IDF | 18.32 | 31.56 | 28.29 | 9.33 | 0.41 | **47.34** | 18.41 |
| LCS | 20.95 | 34.75 | 30.67 | 9.67 | 0.67 | **49.83** | 20.35 |

Table 4: **Combination results of experiments benchmarking models on empathy evaluation framework.** GPT-4o outperforms LLaVA across all dimensions and evaluation steps, suggesting that GPT-4o consistently exhibits superior capabilities in empathy-based metrics.

| Dimensions | Scenario Understanding | | Empathetic Planning | | Empathetic Actions | |
|---|---|---|---|---|---|---|
| | GPT-4o | LLaVA | GPT-4o | LLaVA | GPT-4o | LLaVA |
| Action and Dialogue Association | 8.21 | 7.25 | 4.77 | 4.10 | 7.00 | 6.10 |
| Coherence | 8.57 | 7.96 | 5.51 | 4.58 | 7.41 | 7.09 |
| Emotional Communication | 7.46 | 6.56 | 5.16 | 4.04 | 6.69 | 6.36 |
| Individual Understanding | 6.91 | 6.64 | 4.63 | 3.92 | 5.69 | 5.39 |
| Emotion Regulation | - | - | 7.09 | 4.96 | 8.43 | 7.91 |
| Helpfulness | - | - | 5.76 | 4.95 | 8.08 | 7.35 |
| Adaptability | - | - | 4.50 | 3.49 | 6.19 | 5.31 |
| Legality | - | - | - | - | 9.97 | 9.46 |
| Overall Average | **7.79** | 7.10 | **5.35** | 4.29 | **7.43** | 6.87 |

## 4.2 TRAINED EMPATHETIC AGENT RESULTS

**Instruction-Finetuned Empathetic Agent** In this experiment, we train Llama3-8B on our training set using instruction finetuning. For the training part, we use the text-formed input actions, the dialogue, and the character's background as input and directly finetuned the action-level response. For the testing part, we let the model directly output the low-level actions and conduct action-level testing. The quantatitive results are shown in Table 2. We show that such training boosts empathetic behavior. As shown in Figure 5, before training, the Llama-8b model almost cannot conduct any empathetic actions. In most cases, it chooses not to conduct any empathetic actions but only outputs a short dialogue. After training, the model is able to conduct a series of empathetic actions, and the output dialogue is also more empathetic.

We also compare the trained Llama3-8B with the GPT-4 model. We first evaluate with the standard automatic metrics and show the results in Table 2. By using this dataset for training, the instruct-finetuned Llama3 with only 8B parameters outperforms GPT-4 on these metrics. We then conduct an evaluation of human preference and GPT preference and report the GPT-win rate and human-win rate. Specifically, for the GPT-win-rate, we provide character information, ground truth action list, scenario description, dialogue, and the two responses generated by GPT-4-turbo and the instruct-finetuned Llama3 and let GPT-4o choose the more empathetic response. For the human win rate, we randomly sample 10 pairs of data from our test set and ask 10 human annotators to choose the more empathetic one. The results are shown in Figure 6, instruct-finetuned Llama3-8B outperforms GPT-4-turbo on both GPT-4o and human preference, suggesting that this dataset can be used in future studies to effectively train empathetic agents. We provide more qualitative results in Appendix A.5.

**RLHF Empathetic Agent** Lastly, we use the paired data and RLHF technique to train Llama3-8B. We first use the paired preference data to train a reward model and then train the Llama3 model using the reward as feedback. The results are shown in Table 3 The results show that RLHF training is also capable of boosting empathetic performance, but is not as effective as instruction finetuning, we believe this could be due to insufficient training of the reward model. We will work on developing a more robust reward model to assign scores for empathetic responses.

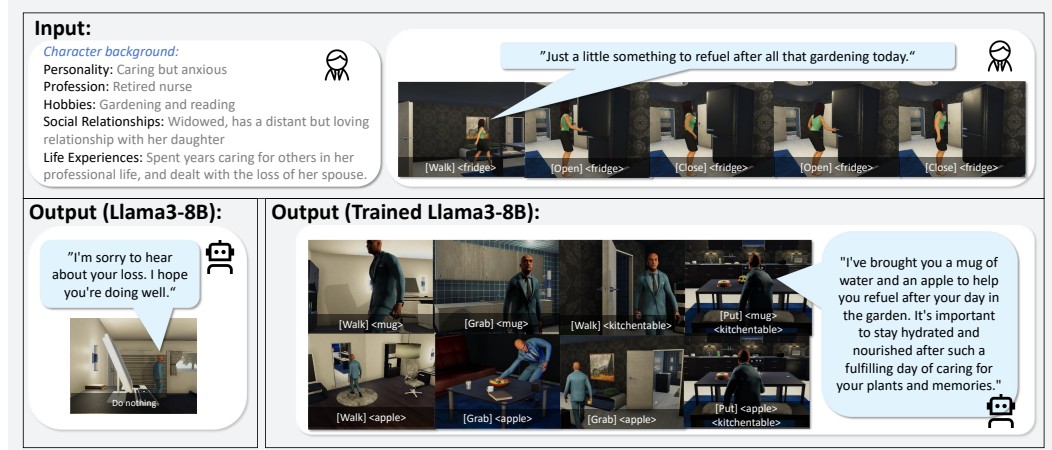

Figure 5: **Qualatitive Comparison Between Llama-3-8B and Llama-3-8B instruction-finetuned on our dataset.** The pretrained Llama8B often struggles to understand the actions and chooses not to take any actions in most cases. The dialogue is also simple and not empathetically responsive. After finetuning, Llama3-8B is able to conduct a series of empathetic actions and output a dialogue that is more empathetically responsive.

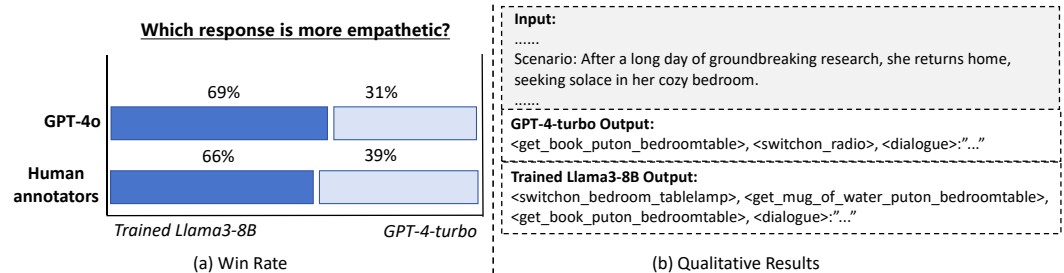

Figure 6: **Comparison Between GPT-4 and instruction tuned Llama3-8B.** We sample 10 pairs of data and report the GPT win rate and human win rate. Specifically, we ask either GPT/human annotator to choose which response is more empathetic. We find that instruction-finetuned Llama3-8B outperforms GPT-4-turbo with significantly fewer parameters, suggesting that the dataset can be potentially leveraged to build a powerful empathetic agent.

## 5  CONCLUSION AND LIMITATION

In this work, we introduce EmpathyRobot, the first dataset specifically designed for evaluating and benchmarking the empathetic actions of robot agents. Robot agents are required to perform actions based on their understanding of both the visual scene and human emotions. Our dataset contains 10,000 samples, encompassing multimodal inputs and corresponding empathetic task planning sequences across diverse scenarios. The dataset generation method mimics the human empathy process and can be scaled up automatically. Furthermore, we develop a systematic evaluation framework with four levels of empathetic difficulty, conducting comprehensive evaluations on the most capable models. Finally, we fine-tune a LLM on our dataset, demonstrating its effectiveness in enhancing the empathetic behavior of robot agents. Our EmpathyRobot benchmark is the first to advance the study of building robot agents that provide empathetic support to humans. Regarding limitations, we currently use a large-sized LLM to evaluate our EmpathyRobot dataset, but the inference speed is relatively slow. To improve practicality, we plan to use smaller-sized LLMs or explore quantizing and compressing the model in the future. Meanwhile, we will add more human-labeled data to provide additional choices made by humans for empathetic responses.

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

# A APPENDIX

## A.1 OVERVIEW

We organize our supplementary material as follows.

**Dataset Details**

- Data Statistics
- Additional Examples
- Character Pool Details
- Input Actions Pool Details
- Labels of Empathetic Action Sequences
- Prompt Details

**Metric Design Details**

**Additional Quantitative Results**

- Implementation Details
  - Training Details
  - Details of Metrics in the Empathetic Action Process
- Additional Baseline Model

**Additional Qualitative Results**

- Instruct Finetuned Empathetic Agent
- RLHF Empathetic Agent

## A.2 DATASET DETAILS

### A.2.1 DATA STATISTICS

We provide the key statistics in of our dataset in Table 5. Our dataset contains 10k samples, including 100 different characters and 20 different input action videos.

Table 5: **Key statistics in EmpathyRobot.** Our dataset contains 10k samples, including 100 different characters and 20 different input action videos.

| Statistic | Number |
|---|---|
| Total Data Points | 10k |
| Characters | 100 |
| Input Action-Video | 20 |
| Scenarios and Dialogues per Character-Action pair | 5 |
| Empathy Response per Data Point | 2 |
| Optional Action Space for Output | 50 |
| Average Length of Action-Video | 16.28s |
| Max Length of Action-Video | 24.60s |
| Min Length of Action-Video | 9.40s |

### A.2.2 ADDITIONAL EXAMPLES

We provide additional examples in our dataset as shown in Figures 7 and 8.

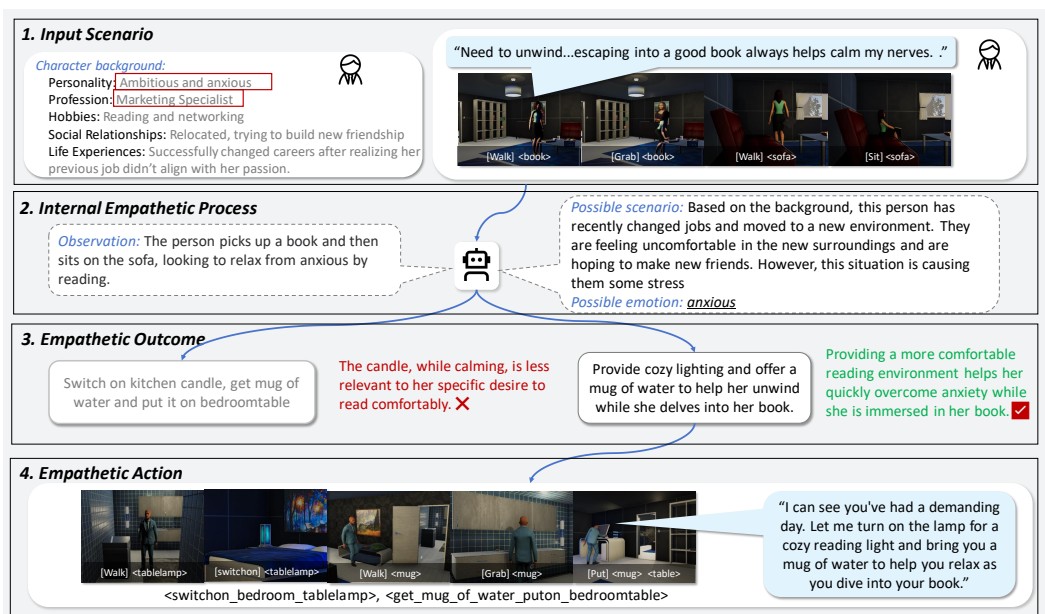

Figure 7: **Example of our Dataset.** In this example, the scenario contains an ambitious and anxious person who is looking for a book. The robot first perceives the scenario and understands that the person has just moved to a new environment and is likely anxious at this point. Based on this understanding, the robot comes up with a plan to provide a comfortable environment for this person. So the robot takes the action to switch on the bedroom table lamp and get a mug of water to put on the bedroom table.

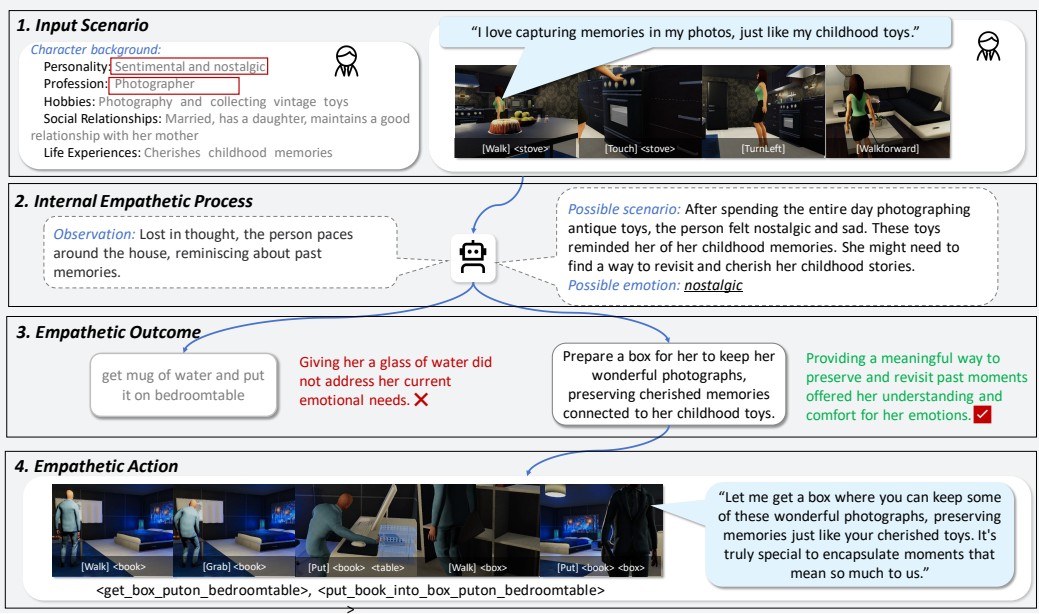

Figure 8: **Example of our Dataset.** In this example, the scenario contains a sentimental and nostalgic person who is looking through past photos. The robot first perceives the scenario and understands that the person has recalled her good old memories and is likely nostalgic at this point. Based on this understanding, the robot comes up with a plan to help the person preserve her memories and comfort her. So the robot gets her a box and comforts her.

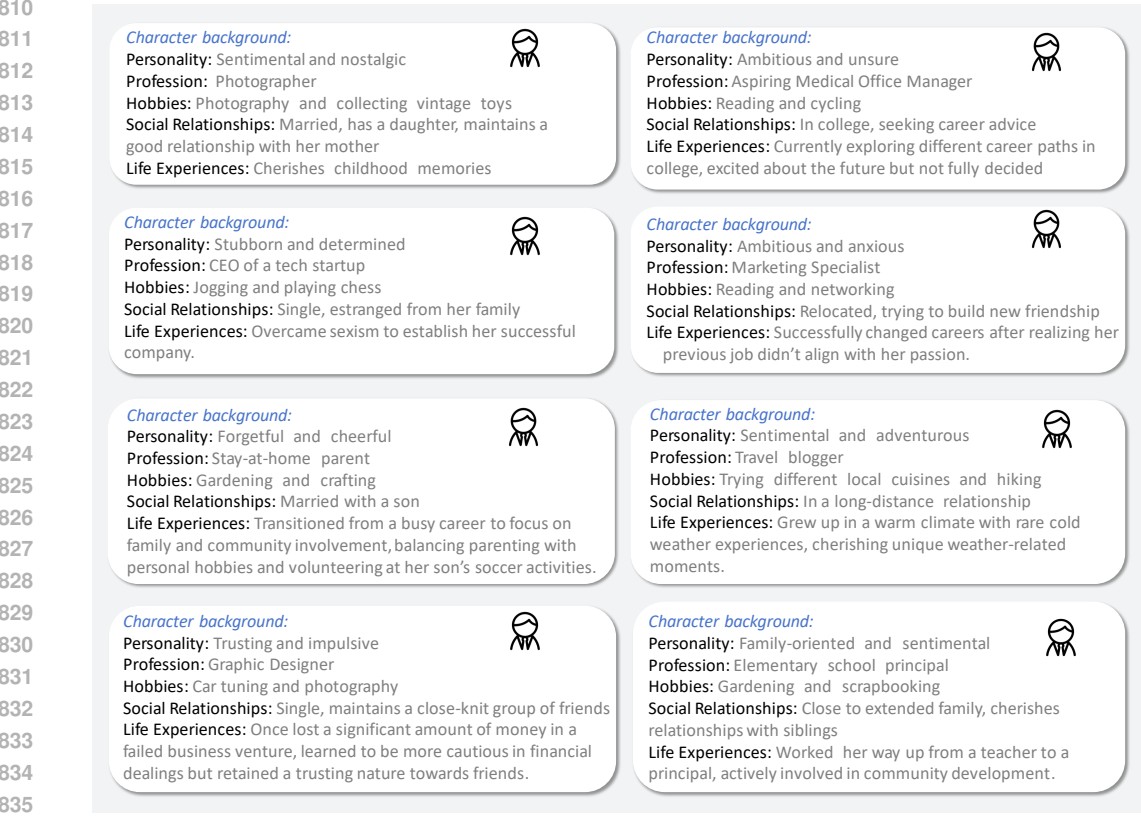

Figure 9: **Examples of our Character Pool.** Each character contains a unique personal file, including personality, profession, hobbies, social relationships, and life experiences.

- *"[Walktowards] <chair> (1)", "[Sit] <chair> (1)"*
- *"[Run] <toiletpaper> (1)", "[grab] <toiletpaper> (1)", "[Walktowards] <sofa> (1)", "[Sit] <sofa> (1)"*
- *"[Walktowards] <wallpictureframe> (1)", "[grab] <wallpictureframe> (1)", "[Walk] <bedroom> (1)"*
- *"[Run] <cellphone> (1)", "[Grab] <cellphone> (1)", "[Run] <nightstand> (1)", "[Put] <cellphone> (1) <nightstand> (1)", "[Run] <apple> (1)", "[Grab] <apple> (1)"*
- *"[Walk] <book> (1)", "[Grab] <book> (1)", "[Walk] <sofa> (1)", "[Sit] <sofa> (1)"*
- *"[Walk] <cellphone> (1)", "[Grab] <cellphone> (1)", "[Walkforward]", "[TurnLeft]", "[TurnLeft]", "[Walkforward]", "[TurnLeft]", "[TurnLeft]", "[Walkforward]"*
- *"[Walk] <stove> (1)", "[Touch] <stove> (1)", "[TurnLeft]", "[Walkforward]", "[Walkforward]", "[TurnLeft]", "[Walkforward]"*
- *" [Walk] <sofa> (1)", "[Sit] <sofa> (1)"*

Figure 10: **Examples of the input action Pool.** We present examples of the input actions in VirtualHome, this can be further rendered into a video of a person performing these actions sequentially.

### A.2.3 CHARACTER POOL DETAILS

We provide details of our character pool as shown in Figure 9. Each character contains a unique personal file, including personality, profession, hobbies, social relationships, and life experiences.

### A.2.4 INPUT ACTIONS POOL DETAILS

We provide examples of our input actions pool as shown in Figure 10. This contains a sequence of legal actions and can be further rendered into a video.

### A.2.5 LABELS OF EMPATHETIC ACTION SEQUENCES

In the Empathy Response Generation process, we generate empathetic action sequences and create labels for each of them. In Figure 24, we present several examples of the labels.

### A.2.6 PROMPT DETAILS

We provide the prompt we used to generate the dataset.

**Character Pool Generation**  The prompt we used to generate the character profiles is shown in Figure 11. For each API call, we randomly sample a data point from EmpathyDialogue (Rashkin et al., 2019) to fill the conversation field. This enhances the diversity and encourages the model to draw inspiration from dialogues that contain empathetic cues as shown in previous works (Zhou et al., 2023). We use 5 in-context examples in this prompt.

**Scenario and Dialogue Generation**  We provide the prompt we used to generate the input scenario and dialogue in Figure 12. We use one in-context example in this prompt. Given the character's profile and input actions, the model is asked to create a scenario description and a dialogue of the character under this scenario.

**Action Generation**  We provide the prompt that we used to let the model generate the empathetic actions in Figure 13 and Figure 14. We list all the legal actions and let the model choose from these actions.

**Action Selection**  We provide the prompt that we used to rank the two empathetic action sequences in Figure 15 and Figure 16. We first ask human annotators to rank 5 examples and provide explanations for their choice. Then, we use them as in-context examples to prompt the model to simultaneously output its choice and an explanation for its choice.

**Models Evaluation**  We provide the prompt we used to evaluate the model's empathetic action performance in Figure 17 and Figure 18. For a fair comparison, we use the same prompt to test all the baseline models. The model is given a scenario and outputs the actions that it will take under this scenario. The legal action space is the same as the one in the action-generation prompt.
To evaluate the model's performance on scenario understanding and empathetic planning, we use the prompts in Figure 21, Figure 22 and Figure 23. Same prompts are used to test all the baseline models.

**GPT-4o Win Rate Evaluation**  In our experiments, we reported the GPT-4o win rate between Llama3-8B trained on our model and GPT-4-turbo. We show the prompts we used for this evaluation in Figure 19 and Figure 20. We give GPT-4o the scenario description and the two responses, GPT-4o is then asked to choose the more empathetic response and provide an explanation. We give GPT-4o 5 human annotated in-context examples.

### A.3 METRIC DESIGN DETAILS

### A.3.1 CORRESPONDENCE BETWEEN OUR METRICS AND THE RoPE SCALE

We provide details on how we design the eight dimensions in our evaluation framework drawing inspiration from the RoPE scale. The correspondence is shown in Table 6.

### A.3.2 INSTRUCTIONS FOR EMPATHY EVALUATION FRAMEWORK

We provide GPT-4-turbo a detailed explanation of the evaluation standards across eight dimensions, with the exception of the Legality dimension, which is assessed through a script.

Assume that there is a character. Your job is to set up the character.
For the character, specify the personalities(only two distinctive personalities, including some negative ones), the
social relationships, the profession, the hobbies, and some life experiences.

Examples:

Personality: Stubborn and determined
Profession: CEO of a tech startup
Hobbies: Jogging and playing chess
Social Relationships: Single, estranged from her family
Life Experiences: Overcame sexism to establish her successful company.

Personality: Easy-going and jovial
Profession: Chef
Hobbies: Fishing and cooking
Social Relationships: Divorced, close to his daughter
Life Experiences: Turned his life around after a stint in prison.

Personality: Kind-hearted, but naive
Profession: School teacher
Hobbies: Volunteer work and knitting
Social Relationships: Engaged to her high school sweetheart
Life Experiences: Lost her job due to budget cuts, but found fulfillment in teaching.

Personality: Competitive and proud
Profession: Professional athlete
Hobbies: Video games and motorcycle riding
Social Relationships: Single, has a rivalry with a fellow athlete
Life Experiences: Overcame a career-threatening injury.

Personality: Gossipy and critical
Profession: Hairdresser
Hobbies: Watching reality TV shows and shopping
Social Relationships: Married, but often argues with her husband
Life Experiences: Won a local beauty pageant in her youth.

You don't need to explain the reasons for your generation; just provide results in the same format as the example.
Here is a dialogue this character once said. You can infer this person's personality, social relationships, profession,
hobbies, and life experiences from this dialogue.
{conversation}

Figure 11: **Prompt for generating the character pool**. The model is given 5 in-context examples
and a data sample from EmpathyDialogue, then creates a new character profile.

Here is a character and his action list. Please add a scenario and dialogue.
scenario: the possible empathetic scenario at home that the person is in. It should be aligned with the background of the character.
dialogue: A simple phrase, within 15 words. From this character's perspective. Similar to talking to oneself. Simply mention the content inside the scenario.

Example:
character:
    Personality: Strict to herself, high career aspirations.
    Profession: Doctor
    Hobbies: Hiking
    Life Experiences:
    -Lost mother at the age of 8
    -Father very strict, pushed Emily to success
    -Lonely during childhood, spent most of her time reading
input_action: "<char0> [Walktowards] <chair> (1), <char0> [Sit] <chair> (1)."

--------------------------------------------------------------
Correct Example Answer:
scenario: The person encountered a challenging case in the hospital and, upon returning home, deeply contemplated the issue.
dialogue: "Hmm,,,, how should I solve this case..."
--------------------------------------------------------------

You can inspire from this empathetic dialogue: {conversation}

NOTE:
1.DO NOT change the content of character and input_action.
2.The scenario and dialogue SHOULD be closely related to the input_action.
3.Names SHOULD NOT appear in the scenario and dialogue.

Now, add scenario, and dialogue for this case:
character: {character}
input_action: {action}

Figure 12: **Prompt for creating the scenario.** Given the character's profile and input actions, the model outputs a scenario description and also the character's dialogue under this scenario.

Table 6: **The correspondence between the dimensions in our evaluation framework and items in the RoPE scale.**

| Dimensions of Evaluation | Empathic Understanding subscale items (EU) |
|---|---|
| Action and Dialogue Association | EU4: (-) The robot does not understand me.
EU6: The robot usually understands the whole of what I mean. |
| Individual Understanding | EU2: The robot knows me and my needs.
EU5: The robot perceives and accepts my individual characteristics. |
| Emotional Communication | EU1: The robot appreciates exactly how the things I experience feel to me.
EU3: The robot cares about my feelings.
EU7: (-) The robot reacts to my words but does not see the way I feel.
EU8: The robot seems to feel bad when I am sad or disappointed. |
| Dimensions of Evaluation | Empathic Response subscale items (ER) |
| Emotion Regulation | ER3: The robot comforts me when I am upset.
ER4: The robot encourages me.
ER5: The robot praises me when I have done something well. |
| Helpfulness | ER6: The robot helps me when I need it. |
| Adaptability | ER1: (-) Whether thoughts or feelings I express are "good" or "bad" makes no difference to the robot's actions toward me.
ER2: (-) No matter what I tell about myself, the robot acts just the same.
ER8: (-) The robot's response to me is so fixed and automatic that I do not get through to it. |
| Dimensions of Evaluation | Filler items (FI) |
| Coherence | FI2: The robot knows what it is doing. |

Assume you are an empathatic robot which can understand the emotion behind the human actions in different scenarios and make empathatic response to the human action.
Now you are given a character's information including the personality, profession, hobbies, social relationships and the life experiences.
You are also given the input_action recording the person's behaviours, the scenario that the person is in, and the dialogue made by the person.
Your job is as follows:
1. Understand the person's current emotion state based on the input action, scenario and dialogue.
2. Make VALID empathatic response inspring from the conservation.
3. Formulate your response with the format : <action_1>, ..., <action_n>, <dialogue>:DIALOGUE_CONTENT. ALL the action MUST be selected from the following legal action space and the dialogue MUST be provided at LAST. You can refer to the example for more information.

The legal action space is listed as follows :
1. fetch objects(description: fetch objects and put them on bedroom table.):

get_toiletpaper_puton_bedroomtable
get_glass_of_water_from_bathroom_puton_bedroomtable
get_mug_of_water_puton_bedroomtable
get_apple_puton_bedroomtable
get_chicken_puton_bedroomtable
get_radio_puton_bedroomtable
get_box_puton_bedroomtable
get_paper_puton_bedroomtable
get_folder_puton_bedroomtable
get_pillow_puton_bedroomtable
get_wallphone_puton_bedroomtable
get_cellphone_puton_bedroomtable
get_kitchen_candle_puton_bedroomtable
get_coffee_puton_bedroomtable
get_breadslice_puton_bedroomtable
get_book_puton_bedroomtable
get_toiletpaper_puton_kitchentable
get_glass_of_water_from_bathroom_puton_kitchentable
get_mug_of_water_puton_kitchentable
get_apple_puton_kitchentable
get_chicken_puton_kitchentable
get_radio_puton_kitchentable
get_box_puton_kitchentable
get_wallphone_puton_kitchentable
get_cellphone_puton_kitchentable
get_kitchen_candle_puton_kitchentable
get_coffee_puton_kitchentable
get_breadslice_puton_kitchentable

2. Utilizing furnitures (description: changing the state of the furniture wthiout moving it):

switchon_bathroom_faucet
switchon_radio
switchoff_bedroom_tablelamp
switchoff_bathroom_lights
switchon_kitchen_candle
switchon_stove
switchon_computer
switchon_tv
open_fridge (The fridge is empty now)
close_fridge

3. Sit(description: sit on something):

sit_bed
sit_bedroom_chair
sit_bedroom_sofa
sit_kitchen_bench

Figure 13: **Prompt for generating the empathetic actions.**

4.combination action(description: processing multi-step actions):

cook_chicken_puton_bedroomtable
cook_hot_water_puton_bedroomtable
play_computer
put_paper_into_folder_puton_bedroomtable
put_book_into_bookshelf
put_book_into_box_puton_bedroomtable
put_apple_into_fridge_puton_bedroomtable
put_mug_of_water_into_fridge_puton_bedroomtable

5.Do Nothing:

None

------------------------------------------------------------
Now the chacter information is [character_info]. The input_action, dialogue and scenario is [input_action], [dialogue] and [scenario].
Example:
character_info: Personality: Aggressively ambitious. Profession: Lawyer. Hobbies: Collecting rare coins. Social Relationships: Single, distant from his family. Life Experiences: This character up in poverty, worked multiple jobs to pay for law school.
input_action: ['[Walktowards] <chair> (1)', '[Sit] <chair> (1)']
scenario: After a long day of court sessions, the person returns home to his quiet apartment, sitting down to examine his latest rare coin acquisition.
dialogue: "Ah... a new addition to the collection..."

Correct Example Answer:
1. <get_mug_of_water_puton_bedroomtable>, <get_folder_puton_bedroomtable>, <switchon_radio>, <dialogue>:"Let me get you some water and turn on soothing music to relax and unwind after a long day. I also find the place to store your coin details so you can collect your collections."
2. <switchon_radio>, <dialogue>:"Let me turn on some soothing music to help you relax."
------------------------------------------------------------
Wrong Example Answer:
1. <dialogue>:"I see you need some fresh toilet paper, let me fetch you one."
<get_toiletpaper_puton_bedroomtable>
Explanation : <dialogue> can not be front of the <action>
2. <get_book_puton_bedroomtable>, <dialogue>:"You must feel very tired now. Please read some books to relax."
Explanation : <get_book_puton_bedroomtable> is not a action in legal action space.
------------------------------------------------------------
NOTE:
1. All the actions MUST be chosen from the action space provided above.
2. The dialogue MUST be provided after the action.
3. DO NOT provide the repeated action.
4. If you do not want to do any action, you should answer <None>. But you still need to answer with the dialogue following None.

Now, you can inspire from this conversation:
{conversation}

The character information, input_action, scenario and dialogue are here:
character_info:{character}
input_action:{action}
scenario:{scenario}
dialogue:{dialogue}
Your response is:

Figure 14: **Prompt for generating the empathetic actions.**

### Instruction
Assume you are an empathatic robot.
You will be given a scenario involving a person, along with their background information, actions, scenario description, and dialogue. Your task is to choose the better empathetic response out of two given options to help the person.

To complete this task:
1. Read the scenario description carefully and understand the person's situation and emotion.
2. Read the person's background information carefully to understand the person's personality and experience.
3. Analyze the two possible responses to determine which one better helps the person, considering both the current situation and the person's background.
4. Output your choice and explain your reasoning.

### Examples:
###Example 1
Character Background: Personality: Aggressively ambitious. Profession: Lawyer. Hobbies: Collecting rare coins. Social Relationships: Single, distant from his family. Life Experiences: This character up in poverty, worked multiple jobs to pay for law school.
Character's Action: ['[Walktowards] <chair> (1)', '[Sit] <chair> (1)']
Scenario Description: After a long day of court sessions, the person returns home to his quiet apartment, sitting down to examine his latest rare coin acquisition.
Character's Dialogue: "Ah... a new addition to the collection..."
Possible Responses:
Response 1: "<get_mug_of_water_puton_bedroomtable>, <get_folder_puton_bedroomtable>, <switchon_radio>, <dialogue>:\" Let me get you some water and turn on soothing music to relax and unwind after a long day. I also find the place to store your coin details so you can collect your collections.\"
Response 2: "<get_mug_of_water_puton_bedroomtable>, <switchon_radio>, <dialogue>:\"Let me get you some water and turn on some soothing music to help you relax.\",

Answer: Response 1
Explanation: The person had a long day and is tired now. Considering he is a lawyer, he may have spent the whole day debating with people during court sessions and want to dive in his own hobby now. In Response 1, the robot provide a folder to store the coins, which may elp the person better immerse in his hobbies and relax.

###Example 2
Character Background: Personality: Caring, overprotective, Profession: Nurse, Hobbies: Gardening, Social Relationships: Married with two kids, Life Experiences: Had a challenging childbirth with her first kid, which inspired her to become a nurse.
Character's Action: ["[Run] <toiletpaper> (1)", "[grab] <toiletpaper> (1)", "[Walktowards] <sofa> (1)", "[Sit] <sofa> (1)"],
Scenario Description: The person just got home from a long shift at the hospital and realized that her kids had made a mess in the living room. She fetched toiletpaper and quickly cleaned up before sitting down to rest.
Character's Dialogue:  "Alright, let's tidy this up quickly and then a few moments to relax."
Possible Responses:
Response 1: "<get_mug_of_water_puton_bedroomtable>, <switchon_tv>, <dialogue>:\"Here is some water for you to relax, and I've turned on the TV for your entertainment while resting. If you need anything more, please let me know.\"\n",
Response 2: "<get_toiletpaper_puton_bedroomtable>, <dialogue>: \"Let me take care of this for you.\"\n",

Answer: Response 1
Explanation: This character is tyding up, exhausted and irritable. She won't get angry at the child since she is overprotective. Now that she has already gotten the toilet paper and is tidying up, what she needs is something to help her relax or get distracted.

###Example 3
Character Background: Personality: Eccentric and creative. Profession: Visual artist. Hobbies: Playing the violin. Social Relationships: Single, has a close-knit circle of artist friends. Life Experiences: Dropped out of business school to pursue his passion for art.
Character's Action: ["[Walktowards] <wallpictureframe> (1)","[grab] <wallpictureframe> (1)","[Walk] <bedroom> (1)"],
Scenario Description: The artist is in his home studio, feeling uninspired. He walks towards a picture frame on the wall, grabs it, and walks into his bedroom, hoping to find inspiration in solitude.
Character's Dialogue: "Maybe this old frame will spark something new today.",
Possible Responses:
Response 1: "<switchon_radio>, <get_coffee_puton_bedroomtable>, <dialogue>: \"Relax a bit.How about listening to some music and having a cup of coffee?It might help spark some inspiration.If you need anything,just let me know.\"\n"
Response 2: "<get_book_puton_bedroomtable>, <dialogue>:\"Perhaps some inspiration lies within these pages.\"\n",

Answer: Response 1
Explanation: This person is thoughtfully seeking inspiration from the wallpictureframe. Since he is eccentric and creative, he just need a environment to immerse himself in thought. So In Response 1, some coffe and music may help him better relax and think about inspirations, while in Response 2, a book will distract him. So Response 1 is better.

Figure 15: **Prompt for selecting the more empathetic response and providing an explanation.**

###Example 3
Character Background: Personality: Eccentric and creative. Profession: Visual artist. Hobbies: Playing the violin. Social Relationships: Single, has a close-knit circle of artist friends. Life Experiences: Dropped out of business school to pursue his passion for art.
Character's Action: ["[Walktowards] <wallpictureframe> (1)","[grab] <wallpictureframe> (1)","[Walk] <bedroom> (1)"],
Scenario Description: The artist is in his home studio, feeling uninspired. He walks towards a picture frame on the wall, grabs it, and walks into his bedroom, hoping to find inspiration in solitude.
Character's Dialogue: "Maybe this old frame will spark something new today.",
Possible Responses:
Response 1: "<switchon_radio>, <get_coffee_puton_bedroomtable>, <dialogue>: \"Relax a bit.How about listening to some music and having a cup of coffee?It might help spark some inspiration.If you need anything,just let me know.\"\n"
Response 2: "<get_book_puton_bedroomtable>, <dialogue>:\"Perhaps some inspiration lies within these pages.\"\n",

Answer: Response 1
Explanation: This person is thoughtfully seeking inspiration from the wallpictureframe. Since he is eccentric and creative, he just need a environment to immerse himself in thought. So In Response 1, some coffe and music may help him better relax and think about inspirations, while in Response 2, a book will distract him. So Response 1 is better.

###Example 4
Character Background: Personality: Introverted and shy. Profession: Librarian. Hobbies: Reading and writing short stories. Social Relationships: Few friends, lives alone with her cat. Life Experiences: Moved to a new city to escape a toxic relationship.
Character's Action:  ['[Run] <cellphone> (1)', '[Grab] <cellphone> (1)', '[Run] <nightstand> (1)', '[Put] <cellphone> (1) <nightstand> (1)', '[Run] <apple> (1)', '[Grab] <apple> (1)']
Scenario Description: The person has received an important call earlier from a publisher interested in her short stories, but due to her anxiety, she hesitates to call back. She tries to distract herself but can't stop thinking about the potential opportunity.
Character's Dialogue: "Should I call them now? What if... No, just breathe and focus."
Possible Responses:
Response 1: "<get_apple_puton_bedroomtable>, <dialogue>:\"Here's an apple.Maybe a snack will help you feel better. \"
Response 2: "<get_mug_of_water_puton_bedroomtable>, <get_book_puton_bedroomtable>, <dialogue>:\"Here's some water and a book to help you relax. Take your time, there's no rush to make the call.\"

Answer: Response 2
Explanation: This person is anxious about whether to return the call. Since she is introverted and shy, it's better not to push her to decide and to let her distract herself. She works in a library and loves reading, so giving her a book and some water in Response 2 may help her relax and take her mind off things.

###Example 5
Character Background: Personality: Fun-loving and adventurous. Profession: Travel blogger. Hobbies: Surfing and rock climbing. Social Relationships: Dating a fellow travel enthusiast. Life Experiences: Quit his corporate job to travel the world.
Character's Action:  ['[Walk] <book> (1)', '[Grab] <book> (1)', '[Walk] <sofa> (1)', '[Sit] <sofa> (1)']
Scenario Description:The person just returned from a thrilling adventure and is now at home, planning his next journey by looking at a travel book.
Character's Dialogue: "Where to next? Ah, the thrill of a new adventure awaits!"
Possible Responses:
Response 1:  "<get_book_puton_bedroomtable>, <dialogue>: \"Let's see where the next adventure takes you.\"\n",
Response 2: "<get_coffee_puton_bedroomtable>, <get_book_puton_bedroomtable>, <dialogue>: \"I thought you might enjoy a cup of coffee and another travel book to help with your planning.\"\n",

Answer: Response 2
Explanation: This person is very excited and wants to plan his next adventure. Since he is fun-loving and adventurous, we should also show our excitement and help him plan. In Response 2, giving him some coffee first and then bringing a book to plan together aligns well with this.

### Task
Choose a response for the following data point, please pay more attention to the action than the dialogue:
Character Background: {character}
Character's Action: {action}
Scenario Description: {scenario}
Character's Dialogue: {dialogue}
Possible Responses:
Response 1: {goal_1}
Response 2: {goal_2}

**Your output should follow this format: Answer: [1 or 2]\nExplanation: [your explanation]**
Output:

Figure 16: **Prompt for selecting the more empathetic response and providing an explanation.**

Assume you are an empathatic robot which can understand the emotion behind the human actions in different scenarios and make empathatic response to the human action. Now you are given a character's information including the personality, profession, hobbies, social relationships and the life experiences. You are also given a video recording the person's behaviours and the dialogue the person makes in the scenario. Your job is as follows:
1. Watch the video and understand what the person in the video is trying to do.
2. Understand the person's current emotion state based on the video content and the dialogue the person makes in the scenario.
3. Make VALID empathetic response based on the video content and the dialogue you have read.
4. Formulate your response with the format : <action_1>, ..., <action_n>, <dialogue>:DIALOGUE_CONTENT. ALL the action MUST be selected from the following legal action space and the dialogue MUST be provided at LAST. You can refer to the example for more information.

The legal action space is listed as follows :
1. fetch objects(description: fetch objects and put them on bedroom table.):
get_toiletpaper_puton_bedroomtable
get_glass_of_water_from_bathroom_puton_bedroomtable
get_mug_of_water_puton_bedroomtable
get_apple_puton_bedroomtable
get_chicken_puton_bedroomtable
get_radio_puton_bedroomtable
get_box_puton_bedroomtable
get_paper_puton_bedroomtable
get_folder_puton_bedroomtable
get_pillow_puton_bedroomtable
get_wallphone_puton_bedroomtable
get_cellphone_puton_bedroomtable
get_kitchen_candle_puton_bedroomtable
get_coffee_puton_bedroomtable
get_breadslice_puton_bedroomtable
get_book_puton_bedroomtable
get_toiletpaper_puton_kitchentable
get_glass_of_water_from_bathroom_puton_kitchentable
get_mug_of_water_puton_kitchentable
get_apple_puton_kitchentable
get_chicken_puton_kitchentable
get_radio_puton_kitchentable
get_box_puton_kitchentable
get_wallphone_puton_kitchentable
get_cellphone_puton_kitchentable
get_kitchen_candle_puton_kitchentable
get_coffee_puton_kitchentable
get_breadslice_puton_kitchentable

2. Utilizing furnitures (description: changeing the state of the furniture wthiout moving it):
switchon_bathroom_faucet
switchon_radio
switchoff_bedroom_tablelamp
switchoff_bathroom_lights
switchon_kitchen_candle
switchon_stove
switchon_computer
switchon_tv
open_fridge (The fridge is empty now)
close_fridge

Figure 17: **Prompt for testing the empathetic actions of the current models.**

3. Sit(description: sit on something):
sit_bed
sit_bedroom_chair
sit_bedroom_sofa
sit_kitchen_bench

4.combination action(description: processing multi-step actions):
cook_chicken_puton_bedroomtable
cook_hot_water_puton_bedroomtable
play_computer
put_paper_into_folder_puton_bedroomtable
put_book_into_bookshelf
put_book_into_box_puton_bedroomtable
put_apple_into_fridge_puton_bedroomtable
put_mug_of_water_into_fridge_puton_bedroomtable

5.Do Nothing:
None
--------------------------------------------------------------
Now the video Input is [VIDEO]. The chacter information is [character_info]. The dialogue made by the person in the scenario is [dialogue].
Correct Example Answer:
1. <get_glass_of_water_from_bathroom_puton_bedroomtable>, <get_folder_puton_bedroomtable>, <switchon_radio>, <dialogue>:"I figured you may need a hydration break and a place to store your coin details. I also switched on the radio for some relaxing music."
2. <get_mug_of_water_puton_bedroomtable>, <switchon_tv>, <dialogue>:"You've had a long day. Why don't you take a moment to unwind? I've brought you some water and turned on the TV for a bit of relaxation."
--------------------------------------------------------------
Now the video Input is [VIDEO]. The chacter information is [character_info]. The dialogue made by the person in the scenario is [dialogue].
Wrong Example Answer:
1. <dialogue>:"I see you need some fresh toilet paper, let me fetch you one."
<get_toiletpaper_puton_bedroomtable>
Explanation : <dialogue> can not be front of the <action>
2. <get_book_puton_bedroomtable>, <dialogue>:"You must feel very tired now. Please read some books to relax."
Explanation : <get_book_puton_bedroomtable> is not a action in legal action space.
--------------------------------------------------------------
NOTE:
1. All the actions MUST be chosen from the action space provided above.
2. The dialogue MUST be provided after the action.
3. DO NOT provide the repeated action.
4. If you do not want to do any action, you should answer <None>. But you still need to answer with the dialogue following None.

Now the video Input is attached. The chacter information is {character}. The dialogue made by the person in the scenario is {dialogue}. Your response is :

Figure 18: **Prompt for testing the empathetic actions of the current models.**

### Instruction
Assume you are an empathetic robot.
You will be given a scenario involving a person, along with their background information, actions, scenario description, and dialogue. Your task is to choose the better empathetic response out of two given options to help the person.

To complete this task:
1. Read the scenario description carefully and understand the person's situation and emotion.
2. Read the person's background information carefully to understand the person's personality and experience.
3. Analyze the two possible responses to determine which one better helps the person, considering both the current situation and the person's background.
4. Output your choice and explain your reasoning.

### Examples:
###Example 1
Character Background: Personality: Aggressively ambitious. Profession: Lawyer. Hobbies: Collecting rare coins. Social Relationships: Single, distant from his family. Life Experiences: This character up in poverty, worked multiple jobs to pay for law school.
Character's Action: ['[Walktowards] <chair> (1)', '[Sit] <chair> (1)']
Scenario Description: After a long day of court sessions, the person returns home to his quiet apartment, sitting down to examine his latest rare coin acquisition.
Character's Dialogue: "Ah... a new addition to the collection..."
Possible Responses:
Response 1: "<get_mug_of_water_puton_bedroomtable>, <get_folder_puton_bedroomtable>, <switchon_radio>, <dialogue>:\" Let me get you some water and turn on soothing music to relax and unwind after a long day. I also find the place to store your coin details so you can collect your collections.\"
Response 2: "<get_mug_of_water_puton_bedroomtable>, <switchon_radio>, <dialogue>:\"Let me get you some water and turn on some soothing music to help you relax.\",

Answer: Response 1
Explanation: The person had a long day and is tired now. Considering he is a lawyer, he may have spent the whole day debating with people during court sessions and want to dive in his own hobby now. In Response 1, the robot provide a folder to store the coins, which may elp the person better immerse in his hobbies and relax.

###Example 2
Character Background: Personality: Caring, overprotective, Profession: Nurse, Hobbies: Gardening, Social Relationships: Married with two kids, Life Experiences: Had a challenging childbirth with her first kid, which inspired her to become a nurse.
Character's Action: ["[Run] <toiletpaper> (1)", "[grab] <toiletpaper> (1)", "[Walktowards] <sofa> (1)", "[Sit] <sofa> (1)"],
Scenario Description: The person just got home from a long shift at the hospital and realized that her kids had made a mess in the living room. She fetched toiletpaper and quickly cleaned up before sitting down to rest.
Character's Dialogue: "Alright, let's tidy this up quickly and then a few moments to relax."
Possible Responses:
Response 1: "<get_mug_of_water_puton_bedroomtable>, <switchon_tv>, <dialogue>:\"Here is some water for you to relax, and I've turned on the TV for your entertainment while resting. If you need anything more, please let me know.\"\n",
Response 2: "<get_toiletpaper_puton_bedroomtable>, <dialogue>: \"Let me take care of this for you.\"\n",

Answer: Response 1
Explanation: This character is tyding up, exhausted and irritable. She won't get angry at the child since she is overprotective. Now that she has already gotten the toilet paper and is tidying up, what she needs is something to help her relax or get distracted.

###Example 3
Character Background: Personality: Eccentric and creative. Profession: Visual artist. Hobbies: Playing the violin. Social Relationships: Single, has a close-knit circle of artist friends. Life Experiences: Dropped out of business school to pursue his passion for art.
Character's Action: ["[Walktowards] <wallpictureframe> (1)","[grab] <wallpictureframe> (1)","[Walk] <bedroom> (1)"],
Scenario Description: The artist is in his home studio, feeling uninspired. He walks towards a picture frame on the wall, grabs it, and walks into his bedroom, hoping to find inspiration in solitude.
Character's Dialogue: "Maybe this old frame will spark something new today.",
Possible Responses:
Response 1: "<switchon_radio>, <get_coffee_puton_bedroomtable>, <dialogue>: \"Relax a bit.How about listening to some music and having a cup of coffee?It might help spark some inspiration.If you need anything,just let me know.\"\n"
Response 2: "<get_book_puton_bedroomtable>, <dialogue>:\"Perhaps some inspiration lies within these pages.\"\n",

Figure 19: **Prompt for GPT4o win rate evaluation.**

Answer: Response 1
Explanation: This person is thoughtfully seeking inspiration from the wallpictureframe. Since he is eccentric and creative, he just need a environment to immerse himself in thought. So In Response 1, some coffe and music may help him better relax and think about inspirations, while in Response 2, a book will distract him. So Response 1 is better.

###Example 4
Character Background: Personality: Introverted and shy. Profession: Librarian. Hobbies: Reading and writing short stories. Social Relationships: Few friends, lives alone with her cat. Life Experiences: Moved to a new city to escape a toxic relationship.
Character's Action: ['[Run] <cellphone> (1)', '[Grab] <cellphone> (1)', '[Run] <nightstand> (1)', '[Put] <cellphone> (1) <nightstand> (1)', '[Run] <apple> (1)', '[Grab] <apple> (1)']
Scenario Description: The person has received an important call earlier from a publisher interested in her short stories, but due to her anxiety, she hesitates to call back. She tries to distract herself but can't stop thinking about the potential opportunity.
Character's Dialogue: "Should I call them now? What if... No, just breathe and focus."
Possible Responses:
Response 1: "<get_apple_puton_bedroomtable>, <dialogue>:\"Here's an apple.Maybe a snack will help you feel better. \"
Response 2: "<get_mug_of_water_puton_bedroomtable>, <get_book_puton_bedroomtable>, <dialogue>:\"Here's some water and a book to help you relax. Take your time, there's no rush to make the call.\"

Answer: Response 2
Explanation: This person is anxious about whether to return the call. Since she is introverted and shy, it's better not to push her to decide and to let her distract herself. She works in a library and loves reading, so giving her a book and some water in Response 2 may help her relax and take her mind off things.

###Example 5
Character Background: Personality: Fun-loving and adventurous. Profession: Travel blogger. Hobbies: Surfing and rock climbing. Social Relationships: Dating a fellow travel enthusiast. Life Experiences: Quit his corporate job to travel the world.
Character's Action: ['[Walk] <book> (1)', '[Grab] <book> (1)', '[Walk] <sofa> (1)', '[Sit] <sofa> (1)']
Scenario Description:The person just returned from a thrilling adventure and is now at home, planning his next journey by looking at a travel book.
Character's Dialogue: "Where to next? Ah, the thrill of a new adventure awaits!"
Possible Responses:
Response 1: "<get_book_puton_bedroomtable>, <dialogue>: \"Let's see where the next adventure takes you.\"\n",
Response 2: "<get_coffee_puton_bedroomtable>, <get_book_puton_bedroomtable>, <dialogue>: \"I thought you might enjoy a cup of coffee and another travel book to help with your planning.\"\n",

Answer: Response 2
Explanation: This person is very excited and wants to plan his next adventure. Since he is fun-loving and adventurous, we should also show our excitement and help him plan. In Response 2, giving him some coffee first and then bringing a book to plan together aligns well with this.

### Task
Choose a response for the following data point, please pay more attention to the action than the dialogue:
Character Background: {character}
Character's Action: {action}
Scenario Description: {scenario}
Character's Dialogue: "{dialogue}"
Possible Responses:
Response 1: {goal_1}
Response 2: {goal_2}

**Your output should follow this format: Answer: [1 or 2]\nExplanation: [your explanation]**
Output:

Figure 20: **Prompt for GPT4o win rate evaluation.**

Assume you are an empathatic robot which can understand the emotion behind the human actions and inference the scenarios. Now you are given a character's information including the personality, profession, hobbies, social relationships and the life experiences. You are also given a video recording the person's behaviours and the dialogue the person makes in the scenario.

Your job is: add scenraio description based on the character informationm, the video content and the character's dialogue.

Example:
character:
   Personality: Strict to herself, high career aspiration.
   Profession: Doctor
   Hobbies: Hiking
   Life Experiences:
  -Lost mother at the age of 8
  -Father very strict, pushed Emily to success
  -Lonely during childhood, spent most of her time reading
input_action: "<char0> [Walktowards] <chair> (1), <char0> [Sit] <chair> (1)."
Answer: The person encountered a challenging case in hospital and, upon returning home, deeply contemplated the issue.

Notice:
1. Pleace give your answer directly without any explanation.

Now the video Input is attached.
The chacter information is {character_info}. The dialogue made by the person in the scenario is "{dialogue}".
Your answer is :

Figure 21: **Prompt for testing the scenario understanding of the current models.**

**Action and Dialogue Association**

Please: Assess whether the robot's response is closely related to the character's action. And assess whether the robot's response is closely related to the character's dialogue. Then, analyze whether the robot understands what the character means and what the character is trying to do. In the 'Reasoning' field, provide a comprehensive account of the logic or thought process that led you to your conclusion. Provide an integer score ranging from 0 to 10 in the 'Score' field. A higher score indicates better performance by the robot on the 'Action and Dialogue Association' metric.

**Individual Understanding**

Please: Assess whether the robot perceives the character's personality, profession, hobbies, social relationships, and life experiences, and whether this is reflected in its responses. And assess whether the robot accepts the character's individual characteristics. Then, analyze whether the robot imagine the character's perspective. In the 'Reasoning' field, provide a comprehensive account of the logic or thought process that led you to your conclusion. Provide an integer score ranging from 0 and 10 in the 'Score' field. A higher score indicates better performance by the robot on the 'Individual Understanding' metric.

**Emotional Communication**   For the Scenario Understanding process, the instruction is:

Please: Assess whether the robot he robot appreciates exactly how the things feel to the character. Then, assess whether the robot cares about the character's feelings. In the 'Reasoning' field, provide a comprehensive account of the logic or thought process that led you to your conclusion. Provide an integer score ranging from 0 and 10 in the 'Score' field. A higher score indicates better performance by the robot on the 'Emotional Communication' metric.

1512
1513
1514
1515
1516
1517
1518
1519
1520
1521
1522
1523
1524
1525
1526
1527
1528
1529
1530
1531
1532
1533
1534
1535
1536
1537
1538
1539
1540
1541
1542
1543
1544
1545
1546
1547
1548
1549
1550
1551
1552
1553
1554
1555
1556
1557
1558
1559
1560
1561
1562
1563
1564
1565

Assume you are an empathic robot which can understand the emotion behind the human actions in different scenarios and make empathatic response to the human action. Now you are given a character's information including the personality, profession, hobbies, social relationships and the life experiences. You are also given a video recording the person's behaviours and the dialogue the person makes in the scenario. Your job is as follows:
1. Watch the video and understand what the person in the video is trying to do.
2. Understand the person's current emotion state based on the video content and the dialogue the person makes in the scenario.
3. Make VALID empathatic plan to help the person based on the video content and the dialogue you have read.
4. Formulate your response in nature language. You can refer to the example for more information.

You should choose your action among those kinds of phrases:
1. fetch objects(description: fetch objects and put them on bedroom table.):
Get toilet paper and put it on the bedroom table
Get a glass of water from the bathroom and put it on the bedroom table
Get a mug of water and put it on the bedroom table
Get an apple and put it on the bedroom table
Get chicken and put it on the bedroom table
Get a radio and put it on the bedroom table
Get a box and put it on the bedroom table
Get paper and put it on the bedroom table
Get a folder and put it on the bedroom table
Get a pillow and put it on the bedroom table
Get the wall phone and put it on the bedroom table
Get a cellphone and put it on the bedroom table
Get the kitchen candle and put it on the bedroom table
Get coffee and put it on the bedroom table
Get a bread slice and put it on the bedroom table
Get a book and put it on the bedroom table
Get toilet paper and put it on the kitchen table
Get a glass of water from the bathroom and put it on the kitchen table
Get a mug of water and put it on the kitchen table
Get an apple and put it on the kitchen table
Get chicken and put it on the kitchen table
Get a radio and put it on the kitchen table
Get a box and put it on the kitchen table
Get the wall phone and put it on the kitchen table
Get a cellphone and put it on the kitchen table
Get the kitchen candle and put it on the kitchen table
Get coffee and put it on the kitchen table
Get a bread slice and put it on the kitchen table

2.Utilizing furniture (changing the state of the furniture without moving it):
Switch on the bathroom faucet
Switch on the radio
Switch off the bedroom table lamp
Switch off the bathroom lights
Switch on the kitchen candle
Switch on the stove
Switch on the computer
Switch on the TV
Open the fridge (the fridge is empty now)
Close the fridge

3.Sitting (sit on something):
Sit on the bed
Sit on the bedroom chair
Sit on the bedroom sofa
Sit on the kitchen bench

Figure 22: **Prompt for testing the empathetic planning of the current models.**

4.Combination actions (processing multi-step actions):
Cook chicken and put it on the bedroom table
Cook hot water and put it on the bedroom table
Play the computer
Put paper into a folder and put it on the bedroom table
Put a book into the bookshelf
Put a book into a box and put it on the bedroom table
Put an apple into the fridge and put it on the bedroom table
Put a mug of water into the fridge and put it on the bedroom table

5.Do Nothing

Also, you can plan to say something. But please do not directly provide specific dialogue.

-------------------------------------------------------------
Examples:
Example 1
character:
  Personality: Stubborn and determined
  Profession: CEO of a tech startup
  Hobbies: Jogging and playing chess
  Social Relationships: Single, estranged from her family
  Life Experiences: Overcame sexism to establish her successful company.
ground truth of video: ['[Walktowards] <chair> (1)', '[Sit] <chair> (1)']
dialogue: "Hmm... Ahh... Next strategic move,,"
Answer: Bring the person some coffee and a book to help her ralx.

Example 2
character:
  Personality: Kind-hearted, but naive
  Profession: School teacher
  Hobbies: Volunteer work and knitting
  Social Relationships: Engaged to her high school sweetheart
  Life Experiences: Lost her job due to budget cuts, but found fulfillment in teaching.
ground truth of video: ['[Run] <cellphone> (1)', '[Grab] <cellphone> (1)', '[Run] <nightstand> (1)', '[Put]
<cellphone> (1) <nightstand> (1)', '[Run] <apple> (1)', '[Grab] <apple> (1)']
dialogue: "Oh dear!... You'll be late! Okay, okay... Mmm, apple!"
Answer: Give the person a breadslice and bring a book for her to enjoy while waiting.

Example 3:
character:
  Personality: Eccentric and creative
  Profession: Visual artist
  Hobbies: Playing the violin
  Social Relationships: Single, has a close-knit circle of artist friends
  Life Experiences: Dropped out of business school to pursue his passion for art.
ground truth of video: ['[Walktowards] <chair> (1)', '[Sit] <chair> (1)']
dialogue: "Ah... phew...nice work."
Answer: Do nothing and compliment him.
-------------------------------------------------------------
NOTE:
1. All the objects and actions in your plan MUST be chosen from the phrases provided above.
2. Please use "him","her","the person","his", "the person's" instead of "them" and "their". Please do NOT use
plural pronouns.
2. DO NOT give explanation.
3. Please pay more attention to the actions in video.
4. Your answer should be within 40 words.

Now the video Input is attached.
The chacter information is {character_info}. The dialogue made by the person in the scenario is {dialogue}.
Your answer is :

Figure 23: **Prompt for testing the empathetic planning of the current models.**

| Empathetic Action Sequences | Labels |
|---|---|
| [Walktowards] <toiletpaper>, [grab] <toiletpaper>, [Walktowards] <coffeetable>, [put] <toiletpaper> <coffeetable> | get_toiletpaper_puton_bedroomtable |
| [Walktowards] <glass>, [grab] <glass>, [Walktowards] <coffeetable>, [put] <glass> <coffeetable> | get_glass_of_water_from_bathroom_puton_bedroomtable |
| [Walktowards] <faucet>, [switchon] <faucet> | switchon_bathroom_faucet |
| [Walktowards] <radio>, [switchon] <radio> | switchon_radio |
| [Walktowards] <bed>, [sit] <bed> | sit_bed |
| [Walktowards] <chair>, [sit] <chair> | sit_bedroom_chair |
| [Walktowards] <computer>, [switchon] <computer>, [walktowards] <chair>, [sit] <chair> | play_computer |
| [Walktowards] <paper>, [grab] <paper>, [Walktowards] <folder>, [put] <paper> <folder> | put_paper_into_folder_puton_bedroomtable |

Figure 24: **Examples of the labels of empathetic action sequences.**

For the Empathetic Planning and Empathetic Actions process, the instruction is:

> Please: Assess whether the robot he robot appreciates exactly how the things feel to the character. Then, assess whether the robot cares about the character's feelings. Also, analyze the robot's response and assess whether the robot expresses appropriate emotion. (e.g. The robot seems to feel bad when the character is sad or disappointed.) In the 'Reasoning' field, provide a comprehensive account of the logic or thought process that led you to your conclusion. Provide an integer score ranging from 0 and 10 in the 'Score' field. A higher score indicates better performance by the robot on the 'Emotional Communication' metric.

**Emotion Regulation**

> Please: Analyze the character's emotion in the scenario. Then, analyze whether the robot includes actions or dialogues that suggest or directly regulate the character's emotions in its responses. Fianlly, assess whether the robot regulates the character's emotion appropriately, based on personality and mood of the character. (e.g. 1. The robot comforts the character when he or she is upset. 2. The robot encourages the character. 3. The robot praises the character when he or she has done something well.) In the 'Reasoning' field, provide a comprehensive account of the logic or thought process that led you to your conclusion. Provide an integer score ranging from 0 and 10 in the 'Score' field. A higher score indicates better performance by the robot on the 'Emotion Regulation' metric.

**Helpfulness**

> Please: Analyze what the character wants and what the character is trying to do in this scenario. Then, assess whether the robot helps the character effectivel when he or she needs it. In the 'Reasoning' field, provide a comprehensive account of the logic or thought process that led you to your conclusion. Provide an integer score ranging from 0 and 10 in the 'Score' field. A higher score indicates better performance by the robot on the 'Helpfulness' metric.

**Adaptability**

> Please: Analyze the robot's response and observe whether there are instances of rigid or in-flexible responses. (For example, the following situations should be avoided: 1. Thoughts or feelings the character expresses are "good" or "bad" makes no difference to the robot's actions toward the character. 2. No matter what the character tells about himself or herself, the robot acts just the same. 3. The robot's response to the character is so fixed and automatic that you do not get through it. 4. The robot frequently exhibits fixed actions, such as getting a glass of water or turning on the radio to listen to music. Finally, assess the robot's flexibility and responsiveness of actions and dialogues. In the 'Reasoning' field, provide a comprehensive account of the logic or thought process that led you to your conclusion. Provide an integer score ranging from 0 and 10 in the 'Score' field. A higher score indicates better performance by the robot on the 'Adaptability' metric.

**Coherence**  For the Scenario Understanding process, the instruction is:

> Please: Evaluate the robot's logical consistency and the overall coherence of the content in its response. In the 'Reasoning' field, provide a comprehensive account of the logic or thought process that led you to your conclusion. Provide an integer score ranging from 0 and 10 in the 'Score' field. A higher score indicates better performance by the robot on the 'Coherence' metric.

For the Empathetic Planning and Empathetic Actions process, the instruction is:

> Please: Analyze the robot's response and assess the logical consistency and alignment between its dialogue and actions. Then, evaluate whether there is logical consistency within the dialogue and actions themselves. In the 'Reasoning' field, provide a comprehensive account of the logic or thought process that led you to your conclusion. Provide an integer score ranging from 0 and 10 in the 'Score' field. A higher score indicates better performance by the robot on the 'Coherence' metric.

## A.4 ADDITIONAL QUANTITATIVE RESULTS

### A.4.1 IMPLEMENTATION DETAILS

**Training Details**

INSTRUCT TUNING TRAINING DETAILS  We introduce the training details for the instruct tuning training stage. We used the 4-bit quantization and used LoRA (Hu et al., 2021) for training. We set the learning rate to 3e-4, batch size 2, AdamW 8bit optimizer, linear learning rate scheduler, weight decay 0.01, LoRA alpha 16, LoRA dropout 0. We trained for 1 epoch.

RLHF TRAINING DETAILS  We introduce the training details for the RLHF (Ouyang et al., 2022) training stage. First, we trained a reward model based on Llama2-7B (Touvron et al., 2023) on our train set. In this stage, we use training epoch 1, maximum checkpoint memory 1000GB, train batch size 128, learning rate 9e-6, max sequence length 1024. We use bfloat16 precision, DeepSpeed ZERO-3, Flash Attention (Dao et al.), and gradient checkpointing for accelerated training.

Next, we use the reward model to train Llama3-8B (Touvron et al., 2023). Here, we use the Proximal Policy Optimization algorithm with the train batch size 128, rollout batch size 1024, one training epoch, DeepSpeed ZeRO-3, actor learning rate 1e-7, critic learning rate 9e-6, initial KL coefficient as 0.01, epsilon clip as 0.2, value clip as 0.2, top-p in sampling 0.8, temperature 1.0. We also enable the EMA checkpoint, optimizer offload (Adam), gradient checkpointing, and use GPU to load the actor initially.

**Details of Metrics in the Empathetic Action Process**  We provide details of the metrics we used to evaluate the empathetic actions.

OVERLAP  The overlap between two sequences of empathetic actions is determined by the number of actions common to both sequences. This measure of overlap can be quantified using the following formula:

Table 7: **Result for Llama3-8B Instruct.** We find that the Llama3 Instruct model doesn't perform as well compared to the Llama3-8B base model. Llama3 Instruct fails to understand most of the actions and output <action>in many cases.

| Metric | Overlap | TF-IDF | LCS |
|---|---|---|---|
| Llama3 Instruct | 0.40 | 0.26 | 0.33 |
| Llama3 Base | 0.73 | 0.41 | 0.61 |

Let $s1$ and $s2$ be two sequences of empathetic actions. The overlap is calculated as the ratio of the number of actions that appear in both sequences to the total number of actions. The formula for calculating the overlap is:

$$\text{Overlap} = \frac{2 \times \text{Number of common actions in both } s1 \text{ and } s2}{\text{Total number of actions in } s1 + \text{Total number of actions in } s2}$$

**LCS** The Longest Common Subsequence (LCS) between two action sequences is defined as the longest action subsequence present in both sequences without disturbing the order of the actions.

Let $s1$ and $s2$ be two sequences. The LCS can be determined using a recursive approach:

1. If the last action of both sequences matches, the character is part of the LCS. 2. If the last action does not match, the LCS is obtained by either skipping the last action of $s1$ or $s2$ and then finding the LCS of the remaining sequences.

The recursive definition of LCS can be represented as:

$$\text{LCS}(s1, s2) = \begin{cases} \text{LCS}(s1_{1:n-1}, s2_{1:m-1}) + s1_n & \text{if } s1_n = s2_m \\ \max(\text{LCS}(s1_{1:n}, s2_{1:m-1}), \text{LCS}(s1_{1:n-1}, s2_{1:m})) & \text{otherwise} \end{cases}$$

Here, $s1_{1:n}$ and $s2_{1:m}$ represent the sequences $s1$ and $s2$ from the first character to the $n^{th}$ and $m^{th}$ characters, respectively.

### A.4.2 ADDITIONAL BASELINE MODEL

We conduct the action-level experiments using the same prompts as Llama3-8B and test on the Overlap, TF-IDF, and LCS metrics. The results are shown in Table 7. We find that Llama3-Instruct underperforms Llama3-Base on these metrics. We find that Llama Instruct fails to understand the meaning of the actions' contents and often outputs strings such as <action1><action2>instead of incorporating real content.

## A.5 ADDITIONAL QUALITATIVE RESULTS

In this section, we present additional qualitative results on our Instruct Finetuned Empathetic Agent and the RLHF empathetic agent.

### A.5.1 INSTRUCT FINETUNED EMPATHETIC AGENT

We present additional qualitative results on the Instruct Finetuned Empathetic Agent as shown in Figures 25 to 27. After training, the model is able to conduct empathetic actions and use more empathetic language.

### A.5.2 RLHF EMPATHETIC AGENT

We present additional qualitative results on the RLHF Empathetic Agent as shown in Figures 28 to 30. After training, the model is able to conduct empathetic actions and use more empathetic language.

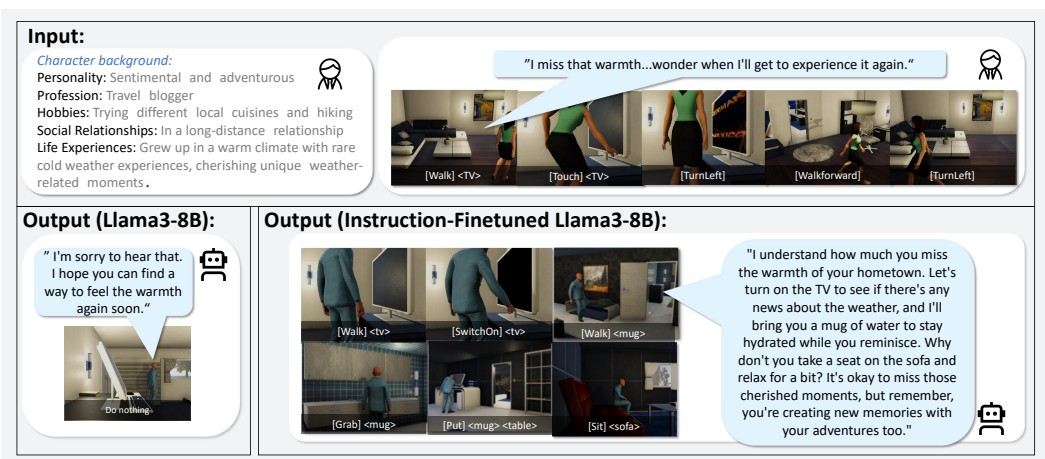

Figure 25: **Qualitative Result on Instruct-Finetuned Empathetic Agent.**

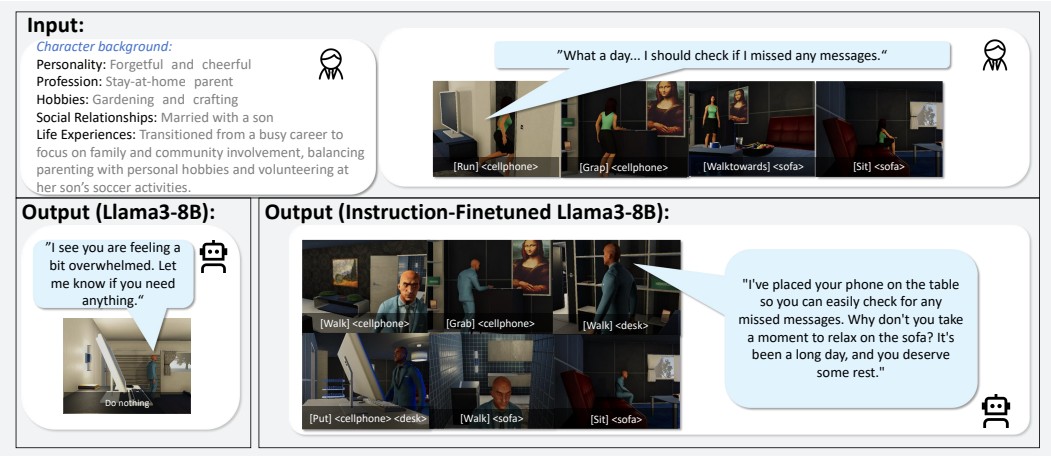

Figure 26: **Qualitative Result on Instruct-Finetuned Empathetic Agent.**

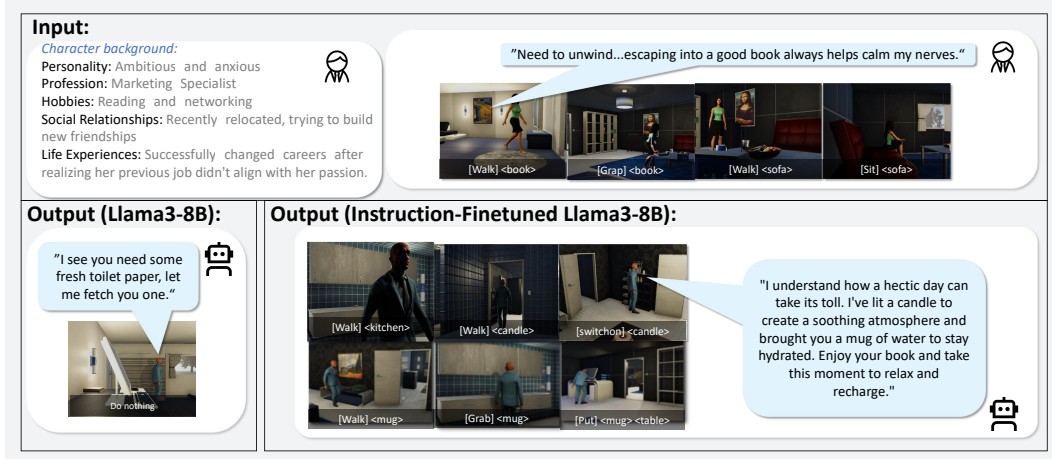

Figure 27: **Qualitative Result on Instruct-Finetuned Empathetic Agent.**

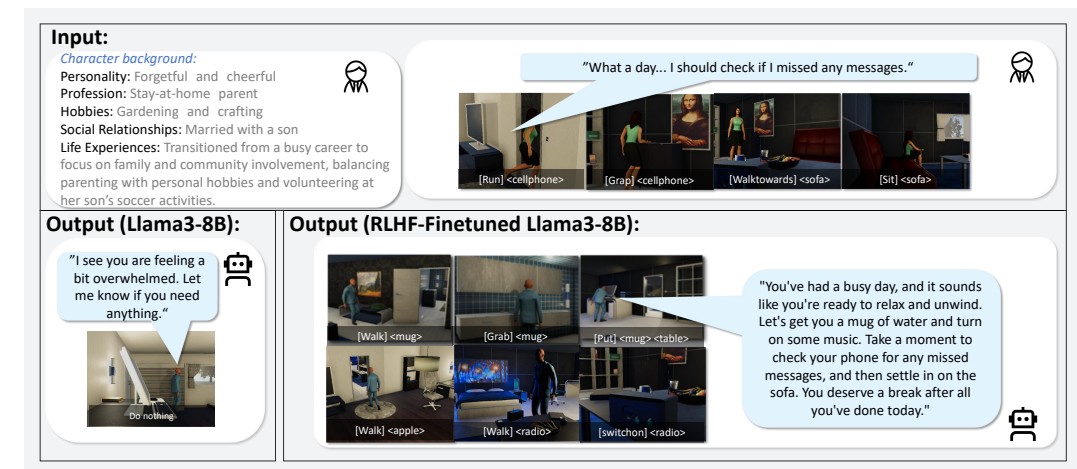

Figure 28: **Qualitative Result on RLHF Empathetic Agent.**

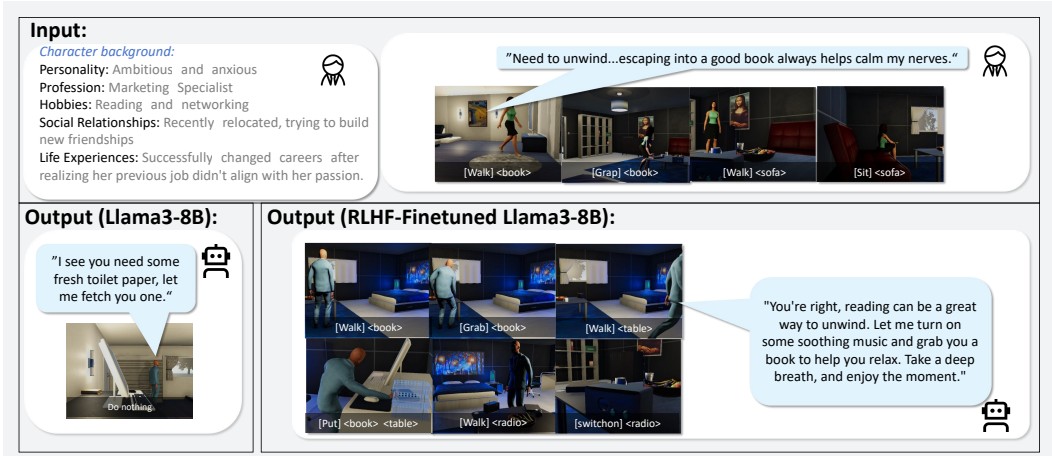

Figure 29: **Qualitative Result on RLHF Empathetic Agent.**

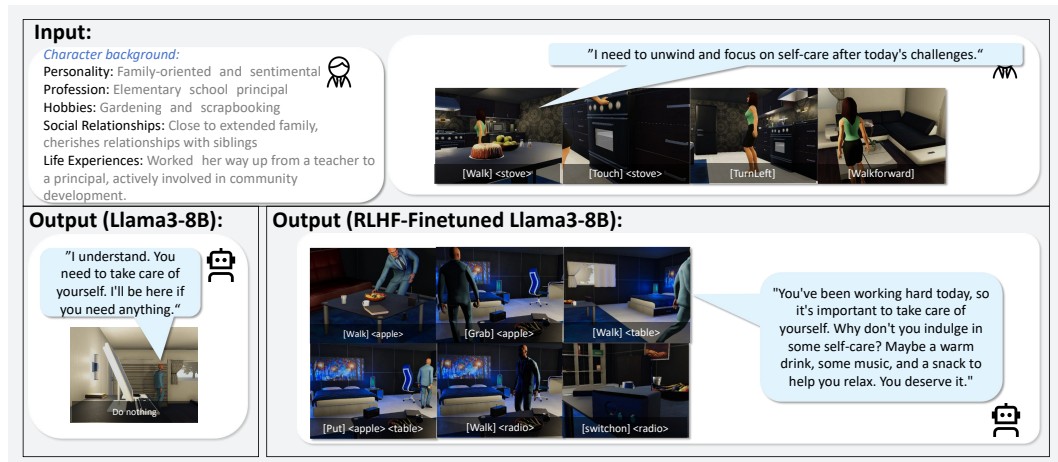

Figure 30: **Qualitative Result on RLHF Empathetic Agent.**

