# OpenReview forum: "EmpathyRobot: A Dataset and Benchmark for Empathetic Task Planning of Robotic Agent"
_ICLR.cc/2025/Conference — Submitted to ICLR 2025_

### Official Review · Reviewer_C66Y · 2024-10-28

**Soundness:** 3
**Presentation:** 4
**Contribution:** 3
**Rating:** 8
**Confidence:** 4

**Summary:**

The paper presents EmpathyRobot, a dataset and benchmark designed to enable robotic agents to exhibit empathetic behaviors by understanding human emotions and planning contextually appropriate actions. It introduces 10,000 multimodal samples and an evaluation framework, aiming to address gaps in existing benchmarks that focus on task completion without assessing empathy. Through experiments with large language models, the authors demonstrate that empathy-driven task planning remains a challenging area for current AI. However, the assumption of “ground truths” for empathy scenarios could be limiting, as empathy may vary significantly based on individual perceptions, making some “correct” responses subjective.

**Strengths:**

- Originality: EmpathyRobot is pioneering in creating a benchmark for empathetic robotic behavior.
- Quality: The dataset is meticulously designed, with a comprehensive evaluation framework.
- Clarity: Clear writing and illustrative examples.
- Significance: This work supports crucial advancements in empathetic AI research, relevant to social intelligence in robotics.

**Weaknesses:**

- The framework’s reliance on “ground truths” for empathy might overlook the subjective nature of empathy, where responses could vary by cultural or personal perspectives.
- Large models face challenges with inference speed; discussions on optimizing these for practical use could strengthen the paper.

**Questions:**

Overall, this paper makes valuable contributions to advancing socially intelligent AI. My question, however, is whether defining two “ground truth” actions per scenario is a reliable measure for performance, given the subjective nature of empathy.

---

> ### Author Response · Authors · 2024-11-23
>
> Thank you very much for your detailed feedback and positive comments! We truly appreciate your recognition of our work. Below, we address your concerns in detail.
> ### **1. Weaknesses 1**
> This is a very worth-thinking problem. First of all, in the current version, the current labeling is participated by people from different backgrounds to mitigate the subject bias. Actually, the mitigation of the influence of subject perspectives is always our primary focus on this work. In the work, we have tried multiple strategies to conduct the experiment and compare their results. These strategies include but are not limited to utilize “golden” rules to give a consensus criteria. For instance, we refer to the Batson et al[1] as a criteria. Additionally, we consider if we need to employ RLHF to fine-tune a large language model, enabling it to assist human annotators in achieving consistent and efficient labeling. The labeling process is also enriched through contextual and cultural calibration, wherein guidelines are periodically refined to reflect cultural variations in empathy perception.
>
> In the future work, we will optimize the labeling process further. We would incorporate consensus-based decision-making among annotators, where two or more reviewers independently evaluate the same data points, with disagreements resolved through structured discussions to ensure balanced and reliable outcomes. To enhance transparency and accountability, annotators are required to provide concise explanations for their labeling decisions, which are subsequently reviewed to identify and mitigate patterns of bias. In the next version, we would place more emphasis on annotating each data point with two or more human-labeled ground truth responses and calculate the evaluation score by averaging the results using multiple ground truths. This allows us to account for diverse personal perspectives on empathy while mitigating potential bias in some ground truth responses.
>
> ### **2. Weaknesses 2**
> The inference performance of the Large Models trained with Empathy Robots has always been our focus. It is the deterministic factor that if the empathy capability could be embedded into the real robots. There are multiple optimizing strategies for the inference of Large Models on edges and it has been verified the high token-density and highly intelligent models could have the excellent real-time inference speed and the memory utilization such as MiniCPM-V2.6[3], Gemini Nano[4], Octopus V2[5]. It is trivial to fine-tune these models with Empathy Robot and enable it with the empathetic inference capabilities. RT-2[1] also demonstrates multiple strategies to optimize the deployment of the large models.
>
> Question One :
> This is a very visionary problem and it is very critical to the quality of our datasets. Two or more annotators would certainly mitigate the subject nature of empathy. Besides, we plan to further refine the labeling process by incorporating consensus-based decision-making among annotators. This approach involves multiple reviewers independently evaluating the same data points, with disagreements systematically resolved through structured discussions to achieve balanced and reliable outcomes. To promote transparency and accountability, annotators will be required to provide succinct justifications for their decisions, which will be reviewed to identify and address potential patterns of bias. Additionally, in the next iteration, we will prioritize annotating each data point with multiple human-labeled ground truth responses. Evaluation scores will then be calculated by averaging these multiple ground truths, enabling us to better account for diverse personal perspectives on empathy while reducing the influence of bias in individual responses.
>
>
> **Reference:**
> [1] Batson, C. D., Lishner, D. A., & Stocks, E. L. (2015). The empathy-altruism hypothesis. The Oxford handbook of prosocial behavior, 259-281.
> [2] Brohan, A., Brown, N., Carbajal, J., Chebotar, Y., Chen, X., Choromanski, K., ... & Zitkovich, B. (2023). Rt-2: Vision-language-action models transfer web knowledge to robotic control. arXiv preprint arXiv:2307.15818.
> [3] Hu, S., Tu, Y., Han, X., He, C., Cui, G., Long, X., ... & Sun, M. (2024). Minicpm: Unveiling the potential of small language models with scalable training strategies. arXiv preprint arXiv:2404.06395.
> [4]Team, G., Anil, R., Borgeaud, S., Alayrac, J. B., Yu, J., Soricut, R., ... & Blanco, L. (2023). Gemini: a family of highly capable multimodal models. arXiv preprint arXiv:2312.11805.
> [5]Chen, W., & Li, Z. (2024). Octopus v2: On-device language model for super agent. arXiv preprint arXiv:2404.01744.

---

> ### Comment · Reviewer_C66Y · 2024-11-26
>
> Thank you for your clarification and responses.
>
> First, I will maintain my ‘accept’ score. I believe this dataset will be a valuable resource for the community.
>
> Regarding the question on the subjective nature of empathy, my concern is **not** about the **consistency** of your “ground truth.” Instead, my worry lies in the potential for **false negatives**—cases where the robot’s behavior is genuinely empathetic but is misclassified as non-empathetic based on the given labels. This could steer approaches evaluated on this dataset toward optimizing for high scores through particular patterns, potentially leading to robotic behaviors that lack diversity and creativity. Such “boring” outcomes would conflict with the initial goal of this paper: fostering engaging and dynamic empathetic interactions.
>
> That said, I don’t have an obvious better solution to offer other than labeling additional data. Therefore, this is intended more as a point for discussion rather than a critique.

---

> > ### Author Response · Authors · 2024-11-29
> >
> > We greatly appreciate your recognition of our work. Regarding the "ground truth", we are making every effort to minimize the potential impact of false negatives.
> >  - Our dataset takes into account the influence of both character and scenario. While empathy does not have an absolute standard (as it varies from person to person), we have constructed our dataset using characters and scenarios from diverse backgrounds. There is a higher likelihood of more appropriate responses for these varied characters and scenarios, which inherently encourages diversity.
> >  - When applied to the real world, our approach can also be customized and learned based on the user's personal background and specific context.
> >  - Furthermore, we do not rely entirely on “ground truth”. In lines 481-485, we use RLHF to train an empathetic agent, which is based on learning from preferences, rather than fixed ground truth labels.
> >
> >
> > Thank you very much for your suggestions on the diversity and creativity of empathetic robotic behaviors. We will carefully explore this aspect in future work.

---

### Official Review · Reviewer_zo8C · 2024-11-03

**Soundness:** 4
**Presentation:** 3
**Contribution:** 4
**Rating:** 8
**Confidence:** 4

**Summary:**

EmpathyRobot is the first dataset for evaluating and enhancing the empathetic actions of robot agents, embedded in the virtual environment and empowered by LLMs.

EmpathyRobot takes realistic social interactions as examples and combines Embodied AI, social events, dialogues, and actions together, making it a comprehensive dataset for the studies on human empathy process.

Besides, EmpathyRobot proposes a systematic evaluation framework with four levels of empathetic difficulty settings, performing comprehensive evaluations on the sota models.

**Strengths:**

The dataset is specifically designed to evaluate and benchmark the empathetic actions of robot agents.

The evaluation is comprehensive, EmpathyRobot designs different levels of "empathy" and conducts comprehensive evaluations on the many models.

The agent design is flexible and can be embedded in a variety of virtual environments.

The methodology of dataset generation makes a lot of sense, giving researchers the opportunity to generate diverse scenarios in different social interactions.

**Weaknesses:**

There is little information about releasing the dataset and how the dataset will be maintained.

What is the practical applications of the dataset. Is there any integration conducted into e.g. character.ai to illustrate the effectiveness of the dataset?

**Questions:**

See weaknesses

---

> ### Author Response · Authors · 2024-11-23
>
> Thank you very much for your feedback and positive comments. You highly praised the flexibility and coherence of our dataset generation pipeline and agents, emphasizing their applicability to various social interactions and virtual environments. We address your concerns below.
>
> ### **Weakness 1**
>
> We have created a homepage for the dataset. Once the paper is accepted, we will publish all relevant information about the dataset on this page, including the number of scenarios, the number of characters, the emotional spectrum, the dataset creation process, benchmarks, and more. Additionally, we will open-source all the pipeline code used for dataset creation, the modified VirtualHome code, and the inference code for all large models used in the benchmarks on platforms such as Hugging Face. Furthermore, we will open-source the code and data for using EmpathyRobot in training real-world robots.
>
> ### **What are the practical applications of the dataset?**
> This is an excellent question that demonstrates great foresight, aligning perfectly with the focus of the next stage of our work. EmpathyRobot prioritizes integrating humanistic factors into robotic research through innovative, high-level methodologies. This approach aims to shift the focus of robotics from purely technical details to addressing how robots can better meet human needs and serve humanity more effectively during their development. Building on this vision, we foresee several key applications for EmpathyRobot in the future:
>
> #### Medical and Healthcare Robots
>
> - By embedding empathy into EmpathyRobot, humanoid robots can move beyond simply considering the feasibility of control in executing tasks to addressing the emotional needs of users.
>
> - Imagine a scenario where a patient feels anxious during a routine medication reminder. An empathetic robot could adjust its tone to be calm and reassuring, offer encouraging words such as, "You're doing great; it's perfectly normal to feel this way," or initiate a conversation about topics the patient enjoys to help ease their anxiety about an upcoming procedure.
>
>
> #### Training Virtual Characters and Digital Humans with Advanced Emotional Perception
>
> - Empathy datasets can be used to train virtual characters or digital humans to respond empathetically to user emotions.
> - For instance, when a user expresses frustration, the digital character learns to respond with supportive and constructive language.
> - Additionally, the dataset incorporates contextual information, such as user interests, cultural references, and prior interactions, enabling digital characters to deliver contextually relevant and personalized responses.
> - This results in enhanced user satisfaction and engagement, empowering digital humans with strong contextual awareness and emotional intelligence.
>
> #### Social Robots for Autism Spectrum Disorder (ASD) Intervention
>
> - Empathy datasets can revolutionize the design of social robots for individuals with autism. These robots, trained on diverse emotional expressions and social interaction scenarios, can act as safe, supportive tools for improving emotional and social skills
>
> Key applications include:
> - **Emotion Recognition and Response Training**: Robots can model and teach appropriate emotional responses using context-specific cues, such as facial expressions, voice tone, or gestures. For instance, when an individual exhibits confusion, the robot could provide clear and patient guidance.
>
> - **Safe Social Practice**: These robots create a nonjudgmental environment where individuals can practice interpreting social signals and developing conversational skills.
>
> #### Empathy-Driven Human-Robot Collaboration in Factories
>
> - Empathy datasets enable the development of industrial robots that prioritize harmonious human-robot collaboration over mere functional efficiency. These robots are designed to understand and adapt to the emotional and physical states of their human coworkers, fostering safer and more productive workplaces.
>
> Key applications include:
> - **Stress and Fatigue Recognition**: Robots equipped with empathy models can detect worker fatigue or stress through biometric data or behavioral patterns, such as slowed movements or irregular task performance. They can then adjust their operation speeds or suggest breaks to prevent accidents
> - **Adaptive Task Allocation**: Robots can dynamically reallocate tasks based on a worker’s emotional or physical condition, ensuring optimal workload distribution while avoiding overburdening team members
> - **Conflict Mitigation**: In team settings, robots can mediate conflicts by recognizing interpersonal tensions and offering neutral, constructive communication to maintain collaboration.

---

### Official Review · Reviewer_vHRs · 2024-11-03

**Soundness:** 2
**Presentation:** 3
**Contribution:** 1
**Rating:** 3
**Confidence:** 5

**Summary:**

This paper proposes a new benchmark called EmpathyRobot to evaluate the "empathetic" actions of agents when interacting with humans in various simulated environments. The authors assess the performance of various rule-based agents on performance on this benchmark.

**Strengths:**

1. The paper proposes a nice benchmark to capture an under-studied element of virtual agents, empathy. The authors derive a clear benchmark with various tasks in VirtualHome to illustrate how one might measure such an ability.

**Weaknesses:**

1. The paper proposes a much more elaborate scenario in the introduction than what is ultimately actually studied in the benchmark: we are only constrained to a specific set of empathy measures and moreover, there is no robotics task or continuous state space in sight (despite the name of the benchmark being EmpathyRobot). The limited discrete rule-based agents tested are a far cry from real deployment scenarios for these types of systems in the real world.

2. There have now been a large body of benchmark-agents-style tasks that, stemming from the Puig et al. VirtualHome environment papers, offer tiny incremental advances on one another. I feel that every other conference, I review a similar paper: "We want to study this [very detailed social/behavioral] element of robots, and so we derive a benchmark from VirtualHome called [x] then evaluate [xyz] agents on it". I do not think these incremental pieces of benchmark work are deserving to be continuously published at high-caliber ML conferences.

**Questions:**

1. How is EmpathyRobot any different from the previous (very long line) of similar social behavior/coordination agents papers? What is the key contribution of novelty that, as a community, we can actually derive from incremental work such as this?

2. How does a highly constrained discrete state and action space actually tell us anything meaningful about robots or robotic behaviors in the real world? Real tasks involving continuous control are a far cry from what the authors write of here, and it would be helpful to have a clear sense of what contribution or task the paper is actually studying.

---

> ### Author Response · Authors · 2024-11-23
>
> Thank you for your review and feedback. Below, we address your concerns.
> ### **Key Contribution:**
> Our work is very different from previous studies of the social behaviors on VirtualHome such as [1] [3] or other LLM-agent frameworks. The main difference is the study of the behavior “empathy”, this is a fundamental human cognitive feature that can be adopted across all settings of social interactions/social simulations. It is a fundamental behavior, not a specific case study.
> ### **1. Questions of the Meaning of Empathy**
> The primary novelty of EmpathyRobot is the focus on empathetic task planning and action generation, which diverges significantly from the task completion benchmarks derived from VirtualHome (e.g., Watch-And-Help [1], Lota-bench [2]). Unlike prior works focusing on general collaborative tasks, EmpathyRobot explicitly evaluates empathetic behavior in scenarios that require agents to process human emotions and then generate contextually appropriate responses.
>
>
> This benchmark advances empathetic robotic behavior by:
> - We introduce the first grounded empathetic action task, evaluated through both established and novel metrics.
> - Presenting a systematic, multi-stage evaluation framework (Scenario Understanding, Empathetic Planning, Empathetic Actions) that is explicitly aligned with cognitive empathy processes on grounded empathetic response.
>
>
> EmpathyRobot does not merely adapt existing tasks but extends their scope by integrating multi-modal emotional understanding, an essential step towards bridging the gap between social perception and emotional intelligence in robots.
>
>
> ### **2. Relevance of Discrete State and Action Spaces**
> We acknowledge the importance of real-world continuous domains for robotics. However, our discrete framework serves as an abstraction layer for isolating and analyzing high-level empathetic cognition. In this work, we provide a pipeline that enables:
> - Clear, interpretable results that directly measure empathetic planning without noise from low-level control challenges.
> - Transferability of insights to continuous domains, as the empathetic reasoning pipeline developed here can guide real-world systems.
>
>
> Additionally, the dataset generation pipeline and evaluation methods are adaptable to simulators with continuous control. Future works can train empathetic robots capable of continuous control or operating in the real world based on the empathetic action sequences and the evaluation metrics that we proposed in this work.
>
>
> ### **3. Contribution on Benchmarking**
> EmpathyRobot fills a gap in existing benchmarks by focusing on empathy as the primary evaluation dimension. This is different from previous social interactions:
> - Evaluation of nuanced empathetic behaviors, such as emotional communication, individual understanding, and adaptability.
> - The introduction of new empathy-specific metrics inspired by psychological constructs, ensuring meaningful evaluation beyond task success rates.
>
>
> This benchmark provides a foundation for advancing empathetic robotic agents in ways that previous works have not systematically addressed. This will push the
>
>
> ### **4. Broader Real-World Implications**
> EmpathyRobot's findings on empathetic task planning have significant implications:
> - Our results highlight challenges in achieving empathy through large-scale models, indicating areas for future research in both cognitive modeling and model training.
> - The emphasis on social contexts prepares robotic systems for applications in healthcare, education, and companionship, where empathetic interaction is essential.
>
>
> While discrete action spaces are limited in scope, they allow for the initial development of empathetic reasoning pipelines, which can later inform design choices for continuous robotic systems.
>
>
> **Reference:**
> [1] Puig, X., Shu, T., Li, S., Wang, Z., Liao, Y. H., Tenenbaum, J. B., ... & Torralba, A. (2020). Watch-and-help: A challenge for social perception and human-ai collaboration. arXiv preprint arXiv:2010.09890.
> [2] Choi, J. W., Yoon, Y., Ong, H., Kim, J., & Jang, M. (2024). Lota-bench: Benchmarking language-oriented task planners for embodied agents. arXiv preprint arXiv:2402.08178.
> [3] Zhang, H., Du, W., Shan, J., Zhou, Q., Du, Y., Tenenbaum, J. B., ... & Gan, C. (2023). Building cooperative embodied agents modularly with large language models. arXiv preprint arXiv:2307.02485.

---

> > ### Author Response · Authors · 2024-11-29
> >
> > Dear Reviewer vHRs,
> > As the discussion phase progresses, we would like to confirm whether our response has addressed your concerns. If you have any remaining questions, we would be happy to discuss and address them. Thank you once again for your valuable feedback.
> > Best, Paper 4375 authors

---

> > > ### Comment · Reviewer_vHRs · 2024-12-02
> > > **Reply to rebuttal**
> > >
> > > Apologies for my late response.
> > >
> > > After reading the author's response to my review, I have decided to keep my score due to 1) remaining fuzziness on the definition of Empathy and therefore its contribution in the literature; 2) the way-too-simplistic action and state spaces for a benchmark in robotics; and 3) the lack of any experiments utilizing the theoretical contributions.

---

> ### Author Response · Authors · 2024-12-03
>
> Thanks for your insightful suggestion. We understand one limitation of our work is the heavy relying on the abstract discrete state on high level neglecting the difficulty of low level robot control.The extensive datasets generated and run on Virtual Home have been widely proven to be deployable in real-world robotic tasks. To further validate the generalization capability of our dataset, we are integrating it with a navigation robot based on the Habitat Lab Simulator. For traditional goal-oriented navigation tasks, we hypothesized specific scenarios to assess the robot's empathy capabilities through its navigation trajectories. Through human intuitive observation, we found that EmpathyRobot can plan safer and more human-friendly routes for robots. We will attach our relevant experiments in the final version. Habitat Lab is a widely used simulator for training robot navigation capabilities, and most robots that demonstrate good navigation performance in this environment have been proven to perform well in real-world scenarios. I believe this can further substantiate that our dataset can significantly contribute to real-world robotic tasks. It is only because our current focus is on high-level empathy design that we had to sacrifice some capabilities in continuous state control of robots during implementation.

---

### Official Review · Reviewer_5aVk · 2024-11-04

**Soundness:** 2
**Presentation:** 3
**Contribution:** 2
**Rating:** 5
**Confidence:** 4

**Summary:**

1) Empathy robot is a large dataset with 10,000 samples for agent actions with the focus on empathetic actions with a three step process: scenario understanding, outcome decision, and action execution.
2) The paper introduces empathy specific metrics motivated by prior works.
3) The paper fine-tunes LLMs on their benchmark. Authors show that their fine-tuned Llama3-8B outperforms strong baselines such as GPT-4o.

**Strengths:**

1) Strong motivation for an empathy-driven dataset
2) Useful empathy specific metrics motivated by prior works

**Weaknesses:**

1) Distinction between emphatic planning vs scenario understanding: the authors claim that scenario understanding is dependent on ‘the person’s underlying emotions’ and empathetic planning includes formulating a high-level plan of what to do after comprehending the scenario. For example, after noticing the person hasn’t eaten anything because of being too upset, the model may come up with a plan like 'Find the person some of his favorite food, then comfort him.’ What is the purpose of the scenario understanding module vs the empathetic planning module when both are conditioned on the person’s emotions? Is it a way of carrying out chain-of-thought? Or are there distinct purposes of the two modules as I don’t quite understand why they can't be combined. Both module seems reasonable but it is difficult to see the importance of each module without ablations e.g., removing the empathic planning module when evaluating empathetic actions.

2) The paper claims that the benchmark helps *evaluate and enhance empathetic actions for robot agents* (Figure 1). I would like to see the usefulness of this on robotics tasks.

3)  In figure 6, what are the prompts for GPT4o? It is difficult to believe that prompt engineering GPT4o would yield to poor empathetic responses.

I would hope that the largest contributions from this paper would be 1) strong evaluations on ablations of the pipeline 2) usefulness on robotics tasks 3) outperforming baselines to show that the benchmark is meaningful. However, with these three points not being well-addressed in the paper, I cannot give a high overall score.

**Questions:**

1) Is the character pool from a ground-truth annotated dataset or is this generated as well?

2) What are labels here?
*Empathy Response Generation Second, we generate empathetic action sequences for each scenario and create labels for them.* (3.2)

---

> ### Author Response · Authors · 2024-11-23
>
> Thank you for your detailed comments. Below, we provide our responses to each of your comments.
> ### **1. Distinction between emphatic planning vs scenario understanding.**
> Emphatic planning and scenario understanding are not **modules** of a pipeline or components of a chain-of-thought [1] process. Instead, they represent distinct **benchmarking levels** for evaluating baseline models. Scenario understanding refers to perceiving a scenario, comprehending the content of the scene, and reasoning about the underlying facts behind it, without the need to respond. In contrast, emphatic planning refers to the emotional understanding of the scenario and then generates a high-level empathy response. For instance, consider a busy CEO who receives an important phone call and quickly grabs an apple. The scenario understanding process should infer that the person likely received an urgent business call, needed a quick snack to satisfy his/her hunger, and was in a rush to get back to work. Meanwhile, the empathic planning process goes a step further—not only recognizing the emotional state in the scenario but also proposing a high-level plan, such as preparing a mug of water and an apple for the person.
> We have updated our paper and the prompts for these two benchmarking stages are presented in Figures 21, 22, and 23 in the Appendix.
> Moreover, as you suggested, we have conducted an ablation study by incorporating the ground truth of the scenario into the input to benchmark the empathetic planning stage. As shown in the table below, adding the ground truth of the scenario to the input improves the performance of GPT-4o and LLaVA in Empathetic Planning. This demonstrates that these models have limitations in scenario understanding, and enhancing their ability in this aspect can help them perform better in the empathetic planning task.
> | Metric | GPT-4o (w.o. gt scenario input) | GPT-4o (w. gt scenario input) | LLaVA (w.o. gt scenario input) | LLaVA (w. gt scenario input) |
> | - | - | - | - | - |
> | BERTScore | 0.622 | **0.634** | 0.576 | 0.593 |
>
> ### **2.  Usefulness on robotics tasks.**
> In our work, we use natural language instructions and visual information as inputs to generate action sequences as outputs. This allows us to focus on the planning ability of different models, helping us better understand how robots might interpret human emotions and decide on appropriate actions, encouraging future advancements in empathetic robotics.
> Additionally, our dataset generation pipeline is highly flexible and adaptable to various simulators, including those supporting continuous domains. In future work, we plan to train robots capable of continuous control or operating in the real world.
>
> ### **3.  Performance of baselines in Figure 6.**
> >In figure 6, what are the prompts for GPT4o? It is difficult to believe that prompt engineering GPT4o would yield to poor empathetic responses.
>
> We did not claim that GPT-4o performs poorly on our benchmark. On the contrary, it outperforms other baselines in the Scenario Understanding and Empathetic Planning stages in Table 1. The prompt for evaluating these two stages of the current models (including GPT-4o) is presented in Figures 21, 22, and 23 in the Appendix. The prompt for evaluating the empathetic actions stage is presented in Figures 17 and 18 in the Appendix.
> In Figure 6, we present the comparison between GPT-4-turbo and our instruction-tuned Llama3-8B, evaluated by GPT-4o and human annotators. We find that instruction-finetuned Llama3-8B outperforms GPT-4-turbo, suggesting that the dataset can be potentially leveraged to build a powerful empathetic agent. The prompt for GPT4o win rate evaluation is presented in Figure 19 and Figure 20 in the Appendix.
>
> ### **4.  Other questions.**
> >Is the character pool from a ground-truth annotated dataset or is this generated as well?
>
> Yes.  The character pool is generated as well, and the prompt for generating is presented in Figure 11 in the Appendix.
> >What are labels here?
>
> We have updated our paper and the labels are presented in Figure 24 in the Appendix.
>
> **Reference:**
> [1] Wei, J., Wang, X., Schuurmans, D., Bosma, M., Xia, F., Chi, E., ... & Zhou, D. (2022). Chain-of-thought prompting elicits reasoning in large language models. Advances in neural information processing systems, 35, 24824-24837.

---

> > ### Comment · Reviewer_5aVk · 2024-11-28
> >
> > I have updated the score as the authors have addressed point 1 and provided some clarifications for points 3 and 4. However, due to the absence of experiments in robotics tasks, I am unable to raise the score any further.

---

> > > ### Author Response · Authors · 2024-12-03
> > >
> > > Thank you for acknowledging our work and rebuttal. We greatly appreciate your decision to upgrade the rating of our paper.
> > > ** For robotics tasks:**
> > > Since our dataset generation pipeline and evaluation methods are highly adaptable, enabling the future development of empathetic robots capable of continuous control or real-world operation, we plan to explore tasks in more continuous spaces in future work.

---

### Meta-Review · Area_Chair_RovG · 2024-12-17

**Metareview:**

The paper received divergent ratings (8,8,5,3). While the reviewers appreciated the value of the benchmark, they initially raised various concerns such as lack of robotics experiments, lack of distinction between emphatic planning vs scenario understanding, and the framework's dependence on "ground truths" for empathy. The authors provided responses to the reviewers that addressed some of the concerns (details below), but there was still no consensus among the reviewers. The AC checked the paper, the reviews and the responses. The AC believes the work is valuable and studies a relatively unexplored problem. However, the AC agrees with reviewers 5aVk and vHRs that the paper requires robotics experiments, either in simulation or the real world, with realistic action and state spaces to make a meaningful contribution. Also, the evaluation metrics are not ideal. They are similar to image captioning metrics which have several issues (human performance is usually low). Due to these issues, rejection is recommended.

**Additional Comments On Reviewer Discussion:**

Reviewer 5aVk updated their score since the response addressed some of the concerns. However, they were still concerned about the absence of experiments in robotics tasks. Reviewer vHRs decided to keep their score due to “1) remaining fuzziness on the definition of Empathy and therefore its contribution in the literature; 2) the way-too-simplistic action and state spaces for a benchmark in robotics; and 3) the lack of any experiments utilizing the theoretical contributions.” Reviewer C66Y leaned towards acceptance, as this dataset provides a valuable resource for developing high-level reasoning algorithms. Reviewer zo8C kept their positive score.

---

### Decision · Program_Chairs · 2025-01-22

Reject